# GLOBAL GRAPH CURVATURE

## ABSTRACT

Recently, non-Euclidean spaces became popular for embedding structured data. However, determining suitable geometry and, in particular, curvature for a given dataset is still an open problem. In this paper, we define a notion of global graph curvature, specifically catered to the problem of embedding graphs, and analyze the problem of estimating this curvature using only graph-based characteristics (without actual embedding the graph). We show that optimal curvature essentially depends on the dimensionality of the embedding space and loss function one aims to minimize via embedding. We review existing notions of local curvature (e.g., Ollivier-Ricci curvature) and analyze their properties theoretically and empirically. In particular, we show that such curvatures are often unable to properly estimate the global one. We also make an important observation: larger dimensionality usually leads to less negative optimal curvature. We propose a new estimator of global graph curvature which, in contrast to the existing ones, reflects this property as it is dimension-specific. We demonstrate that for a given dimension the new estimator is able to distinguish between "hyperbolic" and "Euclidean" networks.

## 1 INTRODUCTION

Representation learning is an important tool for learning from structured data such as graphs or texts (Grover & Leskovec, 2016; Perozzi et al., 2014; Mikolov et al., 2013). State-of-the-art algorithms typically use Euclidean space for embedding. Recently, however, it was found that hyperbolic spaces demonstrate superior performance for various tasks (Nickel & Kiela, 2018; Sala et al., 2018), while in some cases spherical spaces can be useful (Liu et al., 2017). A key characteristic classifying the above-mentioned spaces is curvature, which is negative for hyperbolic spaces, zero for Euclidean and positive for spherical spaces. These findings, therefore, show that certain graphs are better represented in spaces with non-zero curvature. While some methods simply fix curvature (e.g., -1 for hyperbolic space) and then find the optimal embedding of the graph in the corresponding space (Nickel & Kiela, 2018), others try to learn the right curvature and embedding simultaneously (Gu et al., 2019).

In this paper, we analyze the problem of determining a graph curvature suitable for embedding *without* actual embedding the graph itself. The aim is to use some simple graph characteristics to understand whether it is reasonable to embed a graph into some non-Euclidean space and if yes, then which curvature to choose. Having such an estimator would save computational resources, since learning an embedding is computationally expensive and should be done for each space separately. More importantly, it can also save human resources by showing whether it is worth investing resources in the implementation of a more complex embedding algorithm.

We make an important first step in this direction by introducing a concept of *global* graph curvature, which depends on both the dimension and loss function used for the embedding. We consider two loss functions: *distortion*, which is widely used in embedding literature, and *threshold-based* loss (see Section 2), which is more suitable for some practical applications. We demonstrate, both theoretically and empirically, that these loss functions lead to fundamentally different graph curvatures. We also compare several estimators of the global curvature, in particular, the ones based on well-known Ollivier-Ricci and Forman-Ricci *local* graph curvatures. Our analysis shows that all these notions give curvatures that are far from the optimal curvature for embedding the graph (in particular, because they are dimension-independent). This raises the important question of how to properly estimate the best curvature for embedding graphs. We approach this problem by introduc-

ing a new curvature estimator which, in contrast to all existing ones, is dimension-specific. This estimator agrees with our theoretical results and is also able to distinguish between "hyperbolic" and "Euclidean" networks.

One important insight, which we gained based on both theoretical and empirical analysis, is the fact that the optimal curvature is usually smaller for smaller dimensions. In other words, a network which seems to be negatively curved (for some small dimension) usually becomes more neutral as dimensionality grows.

Finally, while in the current research we assume that the global graph curvature is constant, this concept can be extended in the future to cover, e.g., the product spaces from (Gu et al., 2019). We focused on the constant curvature since, first, such spaces are widely used in various applications (Tifrea et al., 2018) and, second, even for constant curvature we observed many non-trivial theoretical and empirical results. These results and insights can eventually be extrapolated to analyze curvature in other, more complicated spaces. To the best of our knowledge, the problem of analyzing different notions of curvature for embedding graphs (even in simple spaces) was not considered before, so we hope that more results will follow.

## 2  BACKGROUND AND RELATED WORK

**Graph embeddings**  For an unweighted connected graph $G = (V, E)$ equipped with the shortest path distance metric $d_G$, a graph embedding $f$ is a map $f : V \to U$, where $U$ is a metric space. We refer to Goyal & Ferrara (2018) for a survey of several graph embedding methods and their applications.

The goal of an embedding is to preserve some structural properties of a graph. Depending on an application, different loss/quality functions are used to measure the quality of a graph embedding. The most well-known is *distortion*:[1]

$$D(f) = \frac{1}{\binom{n}{2}} \sum_{u \neq v} \frac{|d(f(u), f(v)) - d_G(u, v)|}{d_G(u, v)}.$$

Distortion is a global metric, it takes into account all graph distances. However, in some practical applications, it may not be the best choice. For example, in recommendation tasks, we usually deal with a partially observed graph, so a huge graph distance between nodes in the observed part does not necessarily mean that the nodes are not connected in the full graph. Additionally, as discussed in Section 4.3.1, graph distances are hard to preserve: there are simple graphs on just 4 nodes that can be perfectly embedded only in a space of curvature $-\infty$ for any dimension.

Another measure, often used for embeddings, is Mean Average Precision (MAP), which, for a given node $v$, compares the distance-based ranking of other embedded nodes with the graph-neighborhood-based ranking. In contrast to distortion, MAP is scale-invariant, as it cares only about the order. Since changing curvature is equivalent to changing scale, MAP is insensitive to curvature.[2] However, in practical applications like recommendation systems, ranking scores are usually complemented by minimum score thresholds to have personalized sizes of top recommended items. This motivated us to consider another loss function, which is at the same time local and scale-dependent.

Let us define the following class of *threshold-based* loss functions. Given an embedding $f$ of a graph $G$, we (re)construct a graph $G'$ in the following way: $v$ and $u$ are connected in $G'$ iff $d(f(v), f(u)) \leq 1$. Then, any loss function which is based on the comparison of $G$ and $G'$ is called threshold-based. While all our theoretical results hold for *any* threshold-based loss function (see Section 4), in our experiments we measure and compare several functions of this type including, e.g., zero-one loss for the edge classification problem (whether a given node pair is connected or not), see Appendix E.2 for definitions and discussion. Such loss functions are natural in many applications (graph reconstruction, link prediction, recommendations).

---

[1]There are other definitions of distortion in the literature, see, e.g., (Sala et al., 2018).

[2]In other words, it is sufficient to consider only the curvatures -1, 0, 1, corresponding to hyperbolic, Euclidean and spherical spaces. Moreover, by considering a small enough region in hyperbolic or spherical space we get geometry similar to the Euclidean one, so for MAP it is important to distinguish only between -1 and 1.

**Hyperbolic and Spherical Spaces** For many years, Euclidean space was the primary choice for data embeddings (Goyal & Ferrara, 2018). However, it turned out that many observed datasets are well fitted into hyperbolic space (Krioukov et al., 2010). Nickel & Kiela (2017) have shown that such hyperbolic embeddings can improve state-of-the-art quality in several practical tasks, e.g., lexical entailment and link prediction. On the other hand, spherical spaces are also used for embeddings (Liu et al., 2017). Gu et al. (2019) goes even further by suggesting mixed spaces: product manifolds combining multiple copies of spherical, hyperbolic, and Euclidean spaces.

The main advantage of hyperbolic space is that it is "larger": the volume of a ball grows exponentially with its radius. Hence, such spaces are well suited for tree-like structures. On the other hand, spherical spaces are suitable for embedding cycles (Gu et al., 2019). Spherical and hyperbolic spaces are parametrized by *curvature c*, which is positive for spherical space and negative for hyperbolic space. As $c \to 0$, geometry of both these spaces becomes similar to the Euclidean one. We discuss some geometrical properties of these different spaces in Appendix A.

## 3 LOCAL GRAPH CURVATURES

While in this paper we analyze the *global* graph curvature, there are several *local* ones proposed in the literature. Many of them are based on the notion of sectional curvature and Ricci curvature defined for Riemannian manifolds. Intuitively, Ricci curvature controls whether the distance between small spheres is smaller or larger than the distance between the centers of the spheres. Ricci curvature is positive if small spheres are closer than their centers are. We refer to (Jost, 2009; O'neill, 1983) for more details on Ricci curvature.

**Ollivier curvature** Ollivier curvature translates the definition of Ricci curvature to graphs. Again, the idea is to compare the distance between two small balls with the distance between their centers. The distance between balls is defined by the well-known optimal transport distance (a.k.a. Wasserstein distance or earth-mover distance). Formally, for a graph $G$ we consider the shortest path metric on $G$, denoted by $d_G$, and let $W_1^G$ denote the Wasserstein metric with respect to the metric space $(G, d_G)$. Furthermore, for each node $v$ we let $m_v$ denote the uniform probability measure on the neighbors of $v$, i.e., $m_v(u) = \frac{1_{u \sim v}}{\deg(v)}$, where $\deg(v)$ denotes the degree of $v$. Then, the classic definition[3] of Ollivier curvature between two neighboring nodes $v \sim u$ in $G$ is defined as

$$\kappa_G(u, v) = 1 - W_1^G(m_v, m_u). \tag{1}$$

We note that Ollivier curvature always belongs to the interval $[-2, 1]$ (Jost & Liu, 2014).

**Forman curvature** Forman curvature (Sreejith et al., 2016) is based on the discretization of Ricci curvature proposed by Forman (2003). It is defined for a general weighted graph $G$, with both node and edge weights. When the graph $G$ is not weighted, the definition becomes:

$$F_G(u, v) = 4 - (\deg(v) + \deg(u)). \tag{2}$$

Forman was interested in a general discretization of curvature for Riemannian manifolds and his formula includes faces of any dimension. Although this can be translated to graphs (Weber et al., 2017), the formula becomes quite cumbersome. Therefore, in Equation 3 only 1-dimensional faces (edges) are included. One can extend this expression by including higher dimensional faces. This was considered in (Samal et al., 2018), where 2-dimensional faces on three nodes (triangles) were included. In the case of an unweighted graph, we then obtain

$$\hat{F}_G(u, v) = F(u, v) + 3\Delta_{uv} = 4 - \deg(v) - \deg(u) + 3\Delta_{uv}, \tag{3}$$

where $\Delta_{uv}$ is the number of triangles that contain the edge $(u, v)$.

Based on the definitions, both Forman curvatures, especially $F_G(u, v)$, are expected to often be highly negative. Indeed, this is what we observe in Sections 4.3 and 5.

---

[3]Note that Ollivier curvature is defined in much more generality in terms of metrics and random walks, see (Ollivier, 2009). Thus, different version on graphs can be considered. Equation 1 corresponds to the classical choices of graph distance and random walk on the graph.

**Heuristic sectional curvature** A different notion of curvature used by Gu et al. (2019) is based on the following geometric fact. Let $abc$ be a geodesic triangle and let $m$ be the (geodesic) midpoint of $bc$. Then the value $d(a, m)^2 + \frac{d(b,c)^2}{4} - \frac{d(a,b)^2 + d(a,c)^2}{2}$ is equal to zero in euclidean space, is positive in spherical space and negative in hyperbolic space.

For graphs, let $v$ be a node in $G$, $b, c$ neighbors of $v$ and $a$ any other node. Then, we define

$$\xi_G(v; b, c; a) = \frac{1}{2d_G(a, v)} \left( d_G(a, v)^2 + \frac{d_G(b, c)^2}{4} - \frac{d_G(a, b)^2 + d_G(a, c)^2}{2} \right). \tag{4}$$

This resembles the formula above with $m = v$ and the normalization constant $2d_G(v, a)$ is included to yield the right scalings for trees and cycles. To define the graph sectional curvature of a node $v$ and its neighbors $b, c$, we average $\xi_G(v; b, c; a)$ over all possible $a$: $\xi_G(v; b, c) = \frac{1}{|V|-3} \sum_{a \in G \setminus \{v,b,c\}} \xi_G(v; b, c; a)$.[4]

## 4 GLOBAL GRAPH CURVATURE

The problem with these different notions of *local* graph curvature is that they cannot be easily used in practical applications, where data is usually embedded in a space of *constant* curvature. Hence, a *global* notion of curvature is needed. In this section, we propose a practice-driven definition of global graph curvature, discuss how to estimate this curvature based on local notions and compare all curvatures theoretically and empirically for several simple graphs.

### 4.1 DEFINITION

For a graph $G$, let $f(G)$ be an embedding of this graph into a $d$-dimensional space of constant curvature $c$ (spherical, Euclidean or hyperbolic). Assume that we are given a loss function $L(f)$ for the embedding task (see Section 2). Now, let $L_{opt}(G, d, c)$ be the optimal loss for given $d$ and $c$: $L_{opt}(G, d, c) = \min_f L(f)$. Then, we define $d$-dimensional curvature of $G$ in the following way:

$$C_d^L(G) = \arg\min_c L_{opt}(G, d, c). \tag{5}$$

Note that there may be several values of curvature $c$ delivering the minimum of $L_{opt}(G, d, c)$, in this case we say that $C_d^L(G)$ consists of all such points.[5]

Below we analyze global curvatures based on distortion ($C_d^{dist}(G)$) and threshold-based ($C_d^{thr}(G)$) loss functions. In the latter case, in experiments, we measure zero-one loss, but our theoretical results apply to any threshold-based loss, since $L_{opt}(G, d, c)$ reaches its minimum on "perfect" embeddings, where we precisely reconstruct the graph $G$.

### 4.2 APPROXIMATIONS

Let us discuss how local graph curvatures can be used to estimate the global one. In all cases, the standard practice is to average edge or sectional curvature over the graph.

**Ollivier curvature** $\kappa(G) = \frac{1}{|E|} \sum_{u \sim v} \kappa_G(u, v)$.

**Forman curvature** $F(G) = \frac{1}{|E|} \sum_{u \sim v} F_G(u, v)$, $\hat{F}(G) = \frac{1}{|E|} \sum_{u \sim v} \hat{F}_G(u, v)$.

**Average sectional curvature** Let $P_3$ denote the number of paths of length 3 in $G$, then $\xi(G) = \frac{1}{P_3} \sum_{v \in V} \sum_{b < c: b, c \in N(v)} \xi_G(v; b, c)$.

---

[4]We assume that $a$ does not coincide with $b$ or $c$, which do not affect the average much, but makes our results in Section 4.3 more succinct.

[5]Further we slightly abuse notation by writing that $C_d^L(G)$ is a real value if such $c$ is unique and a set of values otherwise.

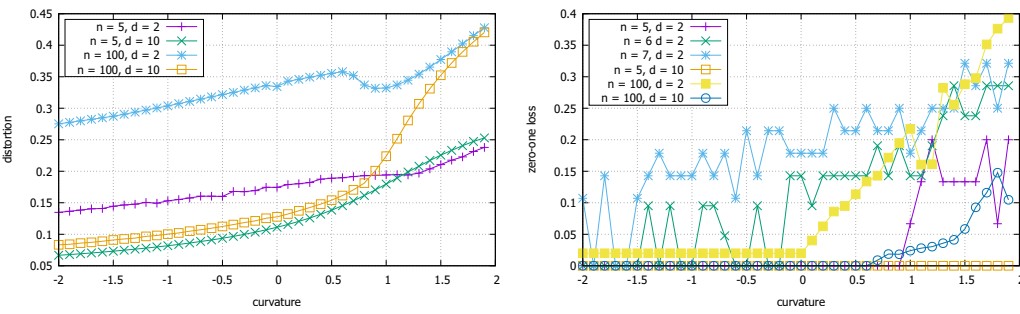

Figure 1: Star $S_n$

It is important to note that all curvatures discussed above do *not* depend on neither dimension $d$ nor loss function $L$. However, as we show below, global curvature defined in Section 4.1 significantly depends on them.

Let us also mention that there is a concept of Gromov's hyperbolicity (Gromov, 1987), which is sometimes used to decide whether it is reasonable to embed a graph to a hyperbolic space. However, first, this estimator cannot be easily converted to curvature, and, second, it does not say anything about the sphericity of data.

### 4.3 THEORETICAL ANALYSIS OF GLOBAL CURVATURE AND ITS APPROXIMATIONS

To better understand the performance of the proposed approximations of global graph curvature, we shall consider several basic graphs and analyze their global curvature and approximations both theoretically and empirically. By studying these graphs we also hope to gain insights into how classic graph topologies influence the curvature of the space in which they can be properly embedded. For more complex graphs, e.g., for combinations of simple graphs studied in this section, it becomes extremely hard to prove rigorous theoretical results. However, the obtained intuitions work in such scenarios, as we illustrate and discuss in Appendix C. The experimental setup for our empirical illustrations is described in Appendix E.3.

#### 4.3.1 STAR $S_n$

It is pointed out in numerous papers that trees are negatively curved. We analyze this theoretically and start with the simplest tree: one central node and $n$ leaves. We denote this graph by $S_n$ and assume that $n \geq 3$.

**Ollivier curvature** Consider any tree graph $T$, let $v, u$ be two neighbors. Then Proposition 2 in (Jost & Liu, 2014) states that

$$\kappa_T(u, v) = -2 \left( 1 - \frac{1}{\deg(v)} - \frac{1}{\deg(u)} \right)^+,  \tag{6}$$

where $t^+ = \max\{0, t\}$. In particular, if either $\deg(v) = 1$ or $\deg(u) = 1$, then $\kappa_T(u, v) = 0$. As a result, for a star we have $\kappa_{S_n}(u, v) = 0$, so $\kappa(S_n) = 0$ and stars are *not* negatively curved according to Ollivier curvature.

**Forman curvature** If follows from Equation 2 and Equation 3 that $F(S_n) = \hat{F}(S_n) = 3 - n$, so stars are highly negatively curved for large $n$ according to Forman curvature.

**Average sectional curvature** Heuristic sectional curvature is defined for a node and its two neighbors. In case of a star we can only take a central node $v$ and two neighboring ones $b$ and $c$. For any other node $a$ we obtain $\xi_{S_n}(v; b, c; a) = -1$. Therefore, by averaging we obtain $\xi(S_n) = -1$.

**Distortion-based curvature** We prove the following theorem (the proof is in Appendix B.3).

**Theorem 4.1.** *Assume that $d$ and $n$ are fixed. If $c$ is bounded below by a constant, then $D(S_n)$ is also bounded below by a constant. If $c \to -\infty$, then the optimal distortion $D_{opt}(S_n) = \Theta\left(\frac{1}{\sqrt{-c}}\right)$. Therefore, for any $S_n$ we have $C_d^{dist}(S_n) = -\infty$.*

This result is illustrated in Figure 6 (left): distortion decreases as curvature becomes smaller. The intuition behind this result is the following: we cannot embed a star $S_3$ with zero distortion into a space of any constant curvature and any dimension, because in case of zero distortion the central node $v$ has to lie on the geodesics between all pairs of leaves, so all 4 nodes have to belong to one geodesics, which is impossible. Moreover, the same problem occurs if any graph $G$ contains $S_3$ as an induced subgraph. On the other hand, if $c \to -\infty$, we can spread all leaves of $S_n$ uniformly on a circle of radius 1 around the central node and distortion of such construction will tend to zero since the distance between the pairs of leaves will tend to 2 (triangles become thinner). Further we will see that if we minimize a threshold-based loss, then any tree can be perfectly embedded with $d = 2$.

**Threshold-based curvature** Here we have the following theorem (the proof is in Appendix B.5).

**Theorem 4.2.** $C_d^{thr}(S_n) = (-\infty, C)$ *for some $C = C(n, d)$, which increases with $d$ and decreases with $n$. In particular, for $d = 2$, if $n < 6$, then $C = \left(\arccos \frac{\cos \frac{2\pi}{n}}{1 - \cos \frac{2\pi}{n}}\right)^2$; if $n = 6$, then $C = 0$; if $n > 6$, then $C = -\left(2 \operatorname{arccosh} \frac{1}{2 \sin \frac{\pi}{n}}\right)^2$.*

The result is illustrated in Figure 6 (right). Note that zero-one loss can be noisy, especially on small graphs, since it is threshold-based, i.e., discrete.

### 4.3.2 TREE $T_b$ WITH BRANCHING FACTOR $b$

We consider a tree $T_b$, $b \geq 2$. For symmetry, assume that the first node has $b + 1$ children, while all other nodes have a parent and $b$ children. Also, for Ollivier, Forman and threshold-based curvatures, we assume that the depth of $T_b$ is infinite. In other cases, our statements hold for any finite tree $T$.

**Ollivier curvature** We can use Equation 6 and get, for any edge, $\kappa_{T_b}(u, v) = -2\left(1 - \frac{1}{b+1} - \frac{1}{b+1}\right) = -2 + \frac{4}{b+1}$. So, $\kappa(T_b) = -2 + \frac{4}{b+1}$, which is negative.

**Forman curvature** It is easy to see that we have $F(T_b) = \hat{F}(T_b) = 4 - 2(b + 1) = 2(1 - b)$.

**Average sectional curvature** In contrast to Ollivier and Forman curvatures, heuristic sectional curvature is global, i.e., it depends on the whole graph, which has to be finite. Note that for any tree, to compute sectional curvature, we average 0 and -1. As a result, for any tree $T$ we have $\xi(T) \in [-1, 0]$ (Gu et al., 2019).

**Distortion-based curvature** On the one hand, our result for $S_n$ implies that if a graph contains $S_3$, then it cannot be embedded with zero distortion in any space. One the other hand, Sarkar (2011) proves that if we scale all edges by a sufficiently large factor $\tau$, then the obtained tree can be embedded to the hyperbolic plane with distortion at most $1 + \varepsilon$ with arbitrary small $\varepsilon$. Note that multiplying graph edges by $\tau$ is equivalent to changing curvature from 1 to $\tau^2$. As a result, Sarkar (2011) proves that we can achieve an arbitrary small distortion if $c \to -\infty$. Hence, $C_d^{dist}(T) = -\infty$ for any $T$.

**Threshold-based curvature** We prove the following theorem (see Appendix B.6).

**Theorem 4.3.** $C_d^{thr}(T_b) = (-\infty, C)$ *for some $C = C(b, d)$, which increases with $d$ and decreases with $b$. The following lower bound holds: $C(b, 2) \geq \left(\frac{2 \log b}{2 \operatorname{arccosh} \frac{\cosh 1}{\cosh 1/2} - 1}\right)^2$.*

Actually, the bound above holds for any tree whose branching is bounded by $b$. Interestingly, while it is often claimed that trees are intrinsically hyperbolic, to the best of our knowledge, we are the first to formally prove that trees can be perfectly embedded in a hyperbolic plane of some curvature.

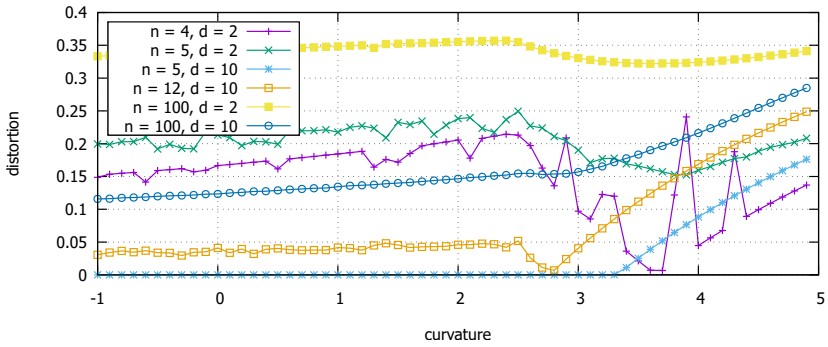

Figure 2: Complete graph $K_n$

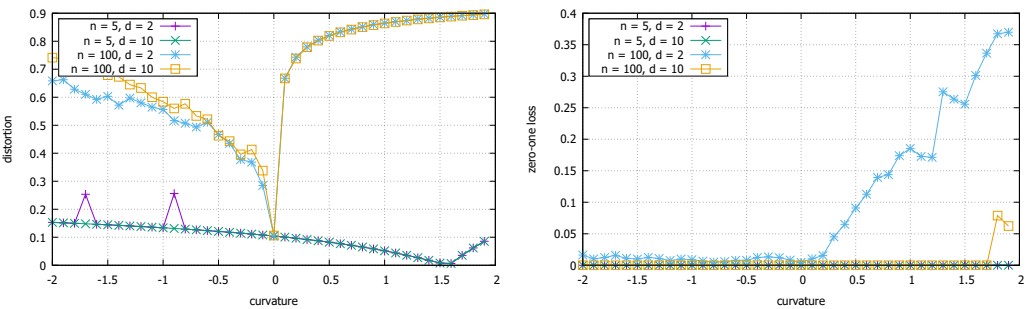

Figure 3: Cycle graph $C_n$

### 4.3.3 COMPLETE GRAPH $K_n$

**Ollivier curvature**   For any two nodes $u$ and $v$, it follows from Example 1 in (Jost & Liu, 2014) that $\kappa_{K_n}(v, u) = \frac{n-2}{n-1}$. Thus, $\kappa(K_n) = \frac{n-2}{n-1}$ and it tends to 1 as $n \to \infty$.

**Forman curvature**   Simple computations yield: $F(K_n) = 6 - 2n$, $\hat{F}(K_n) = n$, i.e., we get either highly positive or highly negative value.

**Average sectional curvature**   It is easy to compute that $\xi(K_n) = \frac{1}{8}$.

**Distortion-based curvature**   The following theorem analyzes $C_d^{dist}(K_n)$ if $d = n - 2$ (see Appendix B.4 for the proof).

**Theorem 4.4.** $C_{n-2}^{dist}(K_n) = \left\{ -\infty, 4\left(\arcsin\sqrt{\frac{n}{2(n-1)}}\right)^2 \right\}$.

Note that the optimal curvature $-\infty$ appears here since embedding of a complete graph can be reduced to embedding of a weighted star. This theorem is illustrated in Figure 2. For $n = 4, d = 2$ we expect to see the minimum at about 3.65, for $n = 12, d = 2$ we expect 2.76, which is indeed the case. For other parameters, we do not have theoretical results, however we see sudden drops in positive curvature. Finally, as expected, distortion decreases as curvature becomes small.

**Threshold-based curvature**   $C_d^{thr}(K_n) = \mathbb{R}$, since we can embed any complete graph perfectly by mapping all nodes to one point.

### 4.3.4 CYCLE GRAPH $C_n$

We consider a cycle $C_n$ with $n \geq 4$.

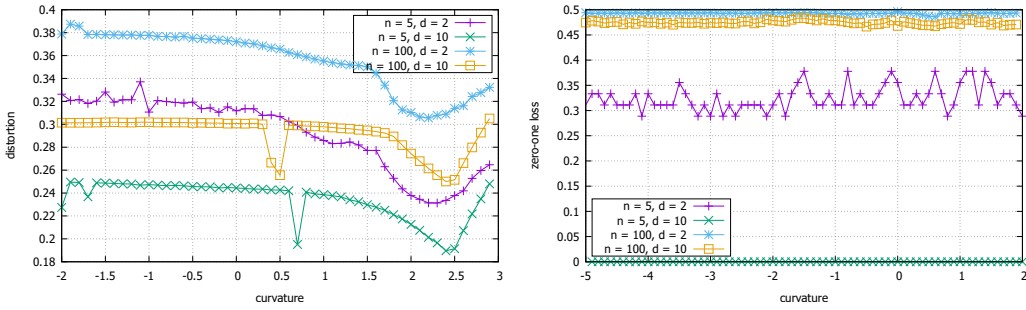

Figure 4: Complete bipartite graphs $K_{n,n}$

**Ollivier curvature**    Let $v \sim u$ be two neighbors. Then it is easy to see that $W_1^G(m_u, m_v) = 1$ and hence $\kappa_G(u, v) = 0$. Thus, $\kappa(C_n) = 0$.

**Forman curvature**    Similarly, it is easy to see that $F(C_n) = \hat{F}(C_n) = 0$.

**Average sectional curvature**    If $n$ is even, then $\xi_{C_n}(v; b, c; a) = 0$ for all points except the one diametrically opposite to $v$ for which we have $\xi_{C_n}(v; b, c; a) = 1$. If $n$ is odd, then for two points we have $\xi_{C_n}(v; b, c; a) = \frac{n}{2(n-1)}$. As a result, $\xi(C_n) = \frac{1}{n-3}$ for even $n$ and $\xi(C_n) = \frac{n}{(n-1)(n-3)}$ for odd $n$.

**Distortion-based curvature**    Here we have that $C_d^{dist}(C_n) = \left(\frac{2\pi}{n}\right)^2$. Indeed, if we consider any three consequent nodes, then the middle one should lie on the geodetic between the other two. So, they all lie on a great circle (of length $n$) from which the result follows. In particular, $C_d^{dist}(C_5) \approx 1.58$, $C_d^{dist}(C_{100}) \approx 0.004$, which is illustrated in Figure 3 (left).

**Threshold-based curvature**    It is easy to see that $C_d^{thr}(C_n) = (-\infty, C)$ with some $C > 0$, which decreases with $n$ and increases with $d$ (see Figure 3, right). A simple lower bound for $C$ is $C \geq \left(\frac{4\pi}{n}\right)^2$, since for such curvature we can embed all nodes into a great circle with distances $1/2$ between the closest ones.

### 4.3.5    COMPLETE BIPARTITE GRAPH $K_{l,m}$

W.l.o.g. we assume that $l \geq m \geq 2$ (the remaining cases are stars and are already considered).

**Ollivier curvature**    We prove the following lemma (the proof is in Appendix B.1).
**Lemma 4.5.** $\kappa(K_{l,m}) = 0$.

**Forman curvature**    $F(K_{l,m}) = \hat{F}(K_{l,m}) = 4 - l - m$.

**Average sectional curvature**    The following lemma holds (see Appendix B.2).
**Lemma 4.6.** $\xi(K_{l,m}) = \frac{-(l-m)^2 + m + l - 2}{(m+l-2)(l+m-3)}$. *In particular, if $m = l$ we get $\xi(K_{l,m}) = \frac{1}{2m-3}$.*

This lemma implies that for balanced complete bipartite graphs $\xi(K_{l,m})$ is positive, but tends to zero as the graph grows.

**Distortion-based curvature**    Figure 4 (left) shows the empirical behavior of distortion depending on curvature. We observe that for all graphs optimal curvature (for the considered range) is between 2 and 2.5, where distortion has a drop. The following proposition gives an intuition about why one could expect this.

**Proposition 4.7.** *For any $d$, $C_d^{dist}(K_{2,2}) = \left(\frac{\pi}{2}\right)^2 \approx 2.47$ and $K_{2,2}$ is the only complete bipartite graph (with at least two nodes in each part) for which zero distortion is achievable.*

Table 1: Compare curvature estimators, distortion

| Dataset/dim | Ollivier | | Forman | | Avg. sectional | | Grad. descent | |
|---|---|---|---|---|---|---|---|---|
| | c | loss | c | loss | c | loss | c | loss |
| Conflict / 2 | -0.2 | **0.229** | -16.5 | 0.261 | 0.25 | 0.269 | 0.0 | **0.229** |
| Conflict / 10 | -0.2 | **0.078** | -16.5 | 0.243 | 0.25 | 0.173 | 0.0 | 0.085 |
| Chicago / 2 | -0.19 | 0.225 | -8.37 | 0.224 | -0.6 | **0.178** | 0.0 | 0.277 |
| Chicago / 10 | -0.19 | 0.045 | -8.37 | 0.216 | -0.6 | **0.024** | 0.0 | 0.080 |
| CSPhDs / 2 | -0.28 | **0.152** | -7.92 | 0.420 | -0.26 | 0.172 | 0.0 | 0.209 |
| CSPhDs / 10 | -0.28 | **0.085** | -7.92 | 0.412 | -0.26 | 0.208 | 0.0 | **0.056** |
| Euroroad / 2 | -0.36 | 0.267 | -1.95 | 0.452 | 0.027 | 0.384 | 0.0 | **0.147** |
| Euroroad / 10 | -0.36 | 0.264 | -1.95 | 0.445 | 0.03 | 0.370 | 0.0 | **0.052** |
| EuroSiS / 2 | -0.17 | 0.247 | -29.1 | **0.197** | 0.27 | 0.298 | 0.0 | 0.263 |
| EuroSiS / 10 | -0.17 | **0.087** | -29.1 | 0.187 | 0.26 | 0.202 | 0.0 | 0.096 |
| Power / 2 | -0.35 | **0.287** | -2.85 | 0.368 | 0.02 | 0.363 | -1.88 | 0.372 |
| Power / 10 | -0.35 | **0.278** | -2.85 | 0.361 | 0.02 | 0.331 | -1.0 | 0.400 |
| Facebook / 2 | 0.308 | 0.290 | -44.7 | 0.327 | 0.153 | 0.240 | 0.0 | 0.238 |
| Facebook / 10 | 0.308 | 0.210 | -44.7 | 0.323 | 0.153 | 0.084 | 0.0 | 0.072 |

Indeed, the result for $K_{2,2}$ follows from the corresponding result on cycle $C_4$. Moreover, if for $K_{l,m}$ we have $l \geq 3$ and $m \geq 2$, then for any two nodes in the part of size $l$ there are at least 2 different geodesics of length 2 between them. Therefore, all such pairs lie at opposite poles of the hypersphere, which is impossible since $l \geq 3$.

**Threshold-based curvature** Threshold-based embeddings to Euclidean spaces are well studied. In particular, it is know that for any graph $G(V, E)$ a perfect threshold embedding exists for some $d \leq |V|$ (Maehara, 1984). However, undirected bipartite graphs are hard to embed in Euclidean space: the bound $d = O(n)$ is known (Maehara, 1984). We empirically observe that they are similarly hard to embed to both spherical and hyperbolic spaces, as shown in Figure 4 (right). Empirically, these graphs are insensitive to curvature (given that it is not too large).

## 4.4 VOLUME-BASED APPROXIMATION

Let us propose a new heuristic curvature estimator which is motivated by the intuition obtained in the previous section as well as by extensive literature arguing that the most important feature of a hyperbolic space is the fact that it has "more volume" in the neighborhood. Therefore, the natural idea is to estimate the volume we need for embedding nodes that are at distance at most $k$ from a given one and then compute the curvature needed to get such volume. We compare the number of nodes at distance exactly $k$ from a node (discounted using the number of edges between them) with the area of a hypersphere of radius $k$ in a space of curvature $c$. The detailed description of the procedure is given in Appendix D. A nice feature of the proposed estimator is that it depends on dimension. Moreover, this dependence is natural: when $d$ becomes larger, curvature becomes smaller in absolute value. As we show in the experiments on various datasets, a similar property is observed for the optimal curvature. Also, it turns out that for a given dimension the new estimator is able to distinguish well between negatively curved and neutral networks. However, despite the potential of this new estimator, the problem of proper global curvature estimation is still wide open.

## 5 EXPERIMENTS

In this section, we compare all discussed curvature estimators empirically. Our experiments are based on a publicly available implementation of graph embedding by Gu et al. (2019).[6] The main advantage of this algorithm is that it works for any curvature: negative, positive or zero. We modified the implementation for our task: in particular, we added a regime responsible for minimizing

---

[6]https://github.com/HazyResearch/hyperbolics

Table 2: Compare curvature estimators, zero-one loss

| Dataset | Ollivier | | Forman | | Avg. sectional | | Grad. descent | |
|---|---|---|---|---|---|---|---|---|
| | c | loss | c | loss | c | loss | c | loss |
| Conflict / 2 | -0.2 | **0.032** | -16.6 | **0.032** | 0.25 | 0.060 | 0.0 | **0.032** |
| Conflict / 10 | -0.2 | 0.001 | -16.6 | 0.025 | 0.25 | $10^{-4}$ | 0.0 | **0.0** |
| Chicago / 2 | -0.19 | **0.004** | -8.37 | 0.005 | -0.6 | **0.004** | 0.0 | **0.002** |
| CSPhDs / 2 | -0.28 | **0.002** | -7.92 | 0.004 | -0.26 | **0.002** | 0.0 | **0.002** |
| Euroroad / 2 | -0.36 | **0.002** | -1.95 | 0.004 | 0.027 | 0.005 | 0.0 | **0.002** |
| EuroSiS / 2 | -0.17 | **0.009** | -29.1 | 0.017 | 0.26 | 0.058 | 0.0 | 0.010 |
| EuroSiS / 10 | -0.17 | **0.003** | -29.1 | 0.012 | 0.26 | **0.003** | 0.0 | 0.007 |
| Power / 2 | -0.35 | **0.002** | -2.85 | 0.006 | 0.02 | 0.006 | 0.0 | **0.002** |
| Facebook / 2 | 0.308 | 0.058 | -44.7 | 0.059 | -11.2 | **0.040** | 0.0 | **0.019** |
| Facebook / 10 | 0.308 | **0.007** | -44.7 | 0.050 | 0.15 | **0.007** | 0.0 | 0.013 |

threshold-based loss functions. The detailed description of our experimental setup is given in Appendix E.3, the datasets and their properties are listed in Appendix E.1.

The results of the comparison for distortion and zero-one loss are presented in Tables 1 and 2, respectively.[7] We also added a method based on the gradient descent, proposed by Gu et al. (2019). This method is used here only as an illustration, since our aim is to compare estimators which do not do actual embedding. Also, the gradient descent algorithm uses the best space based on the value of the loss function, while all other curvature estimators do not use such information. We marked in bold both the global winner and the winner among all curvatures except the gradient descent one. The main goal of this experiment is to show that all known curvatures (Ollivier, Forman and average sectional) have unstable performance and even gradient descent method is not always the best (in both tables). Another important conclusion is that the optimal curvature depends on both dimension and loss function. We show more evidence for that in our extensive analysis in Appendix E.5, where we consider more datasets and more threshold-based loss functions. The experiments also confirm that networks which seems to be negatively curved (for some small dimension) become more neutral as dimensionality grows

In our experiments with the proposed volume-based estimator (see Appendix E.6), we noticed that it is able to predict well whether the hyperbolic space is needed for an embedding or Euclidean space is enough. However, the magnitude for the predicted curvature can be not optimal. Volume-based estimator also predicts that for large dimensions Euclidean space is often a good choice.

# 6 CONCLUSION

In this paper, we introduced a concept of global graph curvature motivated by the practical task of embedding graphs. This curvature depends on the loss function and dimension of a space. To get an intuition about how global graph curvature behaves, we theoretically and empirically analyzed it for several simple graphs. We compared (theoretically and empirically) the global graph curvature and several approximations based on well-known local graph curvatures and showed that they essentially differ. We demonstrated that dimensionality and the choice of a loss function fundamentally affect the global curvature and, in particular, when dimension is larger the optimal curvature usually becomes less negative, which agrees with our theoretical analysis. We also proposed a simple dimension-specific estimator which decides whether it is reasonable to embed into a negatively curved space. Our work shows that the problem of finding the rights space for graph embedding is interesting and non-trivial and we hope our results will encourage further research on global graph curvature.

---

[7]Dimension 10 is not included for zero-one loss if for all curvatures a perfect embedding is possible.

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

## A   GEOMETRICAL PROPERTIES OF SPACES OF CONSTANT CURVATURE

In this section, we recall some useful equalities which will be used throughout the proofs.

In all proofs, we use notation $R$, where $R = \frac{1}{\sqrt{c}}$ in spherical space (corresponds to the radius of a sphere) and in the hyperbolic case $(c < 0)$ $R = \frac{1}{\sqrt{-c}}$ can be considered as a scaling factor compared to the space of curvature 1.

**Law of cosines** Let us consider a triangle with angles $A, B, C$ and the lengths of opposite sides $a, b, c$, respectively.

In Euclidean, space we have:

$$c^2 = a^2 + b^2 - 2\,a\,b\cos C\,.$$

In spherical space, the first law of cosines is:

$$\cos\frac{c}{R} = \cos\frac{a}{R}\cos\frac{b}{R} + \sin\frac{a}{R}\sin\frac{b}{R}\cos C\,,$$

and the second law of cosines is:

$$\cos C = -\cos A\cos B + \sin A\sin B\cos\frac{c}{R}\,.$$

In hyperbolic space, we have

$$\cosh\frac{c}{R} = \cosh\frac{a}{R}\cosh\frac{b}{R} - \sinh\frac{a}{R}\sinh\frac{b}{R}\cos C\,.$$

**Equilateral triangle**

The following equalities follow from the corresponding laws of cosines, assuming that all sides (and angles) are equal.

For hyperbolic space:

$$\cosh\frac{a}{2R} = \frac{1}{2\sin\frac{A}{2}}\,. \tag{7}$$

For spherical space:

$$\cos\frac{a}{R} = \frac{\cos A}{1 - \cos A}\,. \tag{8}$$

**Area and volume of hypersphere** Let $S_d(r)$ and $V_d(r)$ denote area of a hypersphere and volume of a ball of radius $r$ in $d$-dimensional space.

In euclidean space,

$$S_d(r) = d\,C_d\,r^{d-1},$$

$$V_d(r) = C_d\,r^d,$$

where

$$C_d = \frac{\pi^{d/2}}{\Gamma(\frac{d}{2}+1)}\,.$$

In spherical space, sphere of radius $r$ is isometric to Euclidean sphere of radius $R\sin\frac{r}{R}$. Therefore, the area is

$$S_d(r) = d\,C_d\left(R\sin\frac{r}{R}\right)^{d-1},$$

$$V_d(r) = d\,C_d\,R^d\int_0^r\left(\sin\frac{x}{R}\right)^{d-1}dx.$$

Similarly, in hyperbolic space,

$$S_d(r) = d\,C_d\left(R\sinh\frac{r}{R}\right)^{d-1},$$

$$V_d(r) = d\,C_d\,R^d\int_0^r\left(\sinh\frac{x}{R}\right)^{d-1}dx.$$

# B CURVATURE OF SIMPLE GRAPHS

## B.1 PROOF OF LEMMA 4.5 (OLLIVIER CURVATURE FOR BIPARTITE GRAPHS)

Let use denote the node sets in $K_{l,m}$ by $U := \{u_1, \ldots, u_l\}$ and $V := \{v_1, \ldots, v_m\}$. We will prove that for any edge $(u, v)$, $W_1^{K_{l,m}}(m_u, m_v) = 1$, which then implies that $\kappa(K_{l,m}) = 0$. For this we use the dual representation for the Wasserstein distance, see Ollivier (2009) for more details. This states that on the one hand

$$W_1^G(m_u, m_v) = \inf_\rho \sum_{v' \sim v} \sum_{u' \sim u} d_G(v', u') \rho(v', u'),$$

where the infimum is taken over all joint probability measures on the product of the neighborhoods of $v$ and $u$, while on the other hand

$$W_1^G(m_u, m_v) = \sup_f \left( \frac{1}{\deg(v)} \sum_{v' \sim v} f(v') - \frac{1}{\deg(u)} \sum_{u' \sim u} f(u') \right),$$

where the supremum is taken over all 1-Lipschitz functions, i.e $|f(u) - f(v)| \leq d_G(u, v)$.

Note that for any $u \in U, v \in V$ the joint neighborhood is $V \times U$. First we establish an upper bound by considering the product joint probability density on $V \times U$

$$\rho(x, y) = \frac{1}{ml}.$$

It then follows that

$$W_1^{K_{l,m}}(m_u, m_v) \leq \sum_{i=1}^m \sum_{j=1}^l d_G(v_i, u_j) \rho(v_i, u_j) = 1.$$

For the lower bound we define the function

$$f(z) = \begin{cases} 2 & \text{if } z \in V, \\ 1 & \text{if } z \in U. \end{cases}$$

Observe that if $u \in U$ and $v \in V$ then $|f(u) - f(z)| = 1 = d_G(u, v)$. On the other hand, if $u, u' \in U$ then $|f(u) - f(u')| = 0 \leq 2 = d_G(u, u')$ and similar for $v, v' \in V$. Thus we conclude that $f$ is 1-Lipschitz. It now follows that

$$W_1^{K_{l,m}}(m_u, m_v) \geq \frac{1}{m} \sum_{i=1}^m f(v_i) - \frac{1}{l} \sum_{j=1}^l f(u_j) = 1,$$

which completes the proof.

## B.2 PROOF OF LEMMA 4.6 (AVERAGE SECTIONAL CURVATURE FOR BIPARTITE GRAPHS)

If $v$ and $a$ are in the same part of the bipartite graph, then $\xi_{K_{l,m}}(v; b, c; a) = 1$, otherwise $\xi_{K_{l,m}}(v; b, c; a) = -1$. Therefore, if $a$ belongs to the part of size $l$, sectional curvature is $\xi_{K_{l,m}}(v; b, c) = \frac{l-m+1}{l+m-3}$, otherwise it is $\xi_{K_{l,m}}(v; b, c) = \frac{m-l+1}{l+m-3}$. As a result, by averaging over all triplets, we get

$$\xi(K_{l,m}) = \frac{1}{l\binom{m}{2} + m\binom{l}{2}} \left( l\binom{m}{2} \frac{l-m+1}{l+m-3} + m\binom{l}{2} \frac{m-l+1}{l+m-3} \right) = \frac{-(l-m)^2 + m + l - 2}{(m+l-2)(l+m-3)}.$$

## B.3 PROOF OF THEOREM 4.1 (DISTORTION-BASED CURVATURE FOR STARS)

The general idea is the following: we take any curvature $c$ and prove a lower bound on distortion (for any given dimension $d$). Then, we obtain an upper bound on optimal distortion which tends to zero as $c \to -\infty$, which gives the claimed result $C_d^{dist}(S_n) = -\infty$.

First, let us analyze the lower bound on distortion. Recall that distortion of a graph is the average distortion over all pairs of nodes. Let $v$ be the central node and $v_1, \ldots, v_n$ ($n \geq 3$) be its neighbors. Then, for any embedding $f$, we have

$$D(S_n) = \frac{1}{\binom{n+1}{2}} \left( \sum_{v_i} |d(f(v), f(v_i)) - 1| + \sum_{v_i \neq v_j} \frac{|d(f(v_i), f(v_j)) - 2|}{2} \right)$$

$$= \frac{1}{\binom{n+1}{2}} \sum_{1 \leq i_1 < i_2 < i_3 \leq n} \left( \sum_{1 \leq j \leq 3} \frac{|d(f(v_{i_j}), f(v)) - 1|}{\binom{n-1}{2}} + \sum_{1 \leq j < k \leq 3} \frac{|d(f(v_{i_j}), f(v_{i_k})) - 2|}{2(n-2)} \right).$$

Let $D_{min}$ be the minimum value of the following weighted distortion of a star with 3 leaves:

$$D_{min} = \min_f \sum_{1 \leq j \leq 3} \frac{|d(f(v_j), f(v)) - 1|}{(n-1)/4} + \sum_{1 \leq j < k \leq 3} |d(f(v_j), f(v_k)) - 2|,$$

then

$$D(S_n) \geq \frac{\binom{n}{3}}{2(n-2)\binom{n+1}{2}} D_{min} = \frac{(n-1)D_{min}}{6(n+1)}. \tag{9}$$

Hence, it remains to find a lower bound on $D_{min}$, i.e., a lower bound for a weighted distortion of $S_3$ with central node $v$ and three leaves $v_1, v_2, v_3$. If we consider three angles at the node $v$, then at least one of them is $\alpha \leq 2\pi/3$, so we can get a lower bound by only considering this triangle, which is, w.l.o.g., formed by $v, v_1, v_2$.

$$D_{min} \geq |d(f(v_1), f(v_2)) - 2| + \frac{|d(f(v_1), f(v)) - 1| + |d(f(v_2), f(v)) - 1|}{(n-1)/2}.$$

Denote $d(f(v_1), f(v)) = x = 1 + \varepsilon$, $d(f(v_2), f(v)) = y = 1 + \delta$, $d(f(v_1), f(v_2)) = z = 2 + \varepsilon + \delta - \varphi$ with some $\varepsilon, \delta$ and some $\varphi > 0$ (from triangle inequality). Assume that $|\varepsilon| < 1/2$ and $|\delta| < 1/2$ (otherwise the lower bound is trivial). Now we use the law of cosines to get a lower bound on $\varphi$. We consider Euclidean and hyperbolic spaces separately and note that the bound obtained in Euclidean space also holds in spherical spaces (with any $c$).

In Euclidean space, using triangle inequality, we get $\varphi = x + y - z > 0$. So, in Euclidean and spherical spaces $\varphi$ is bounded below by a constant.

In hyperbolic space the law of cosines gives (recall that $R = \frac{1}{\sqrt{-c}}$):

$$\cosh \frac{z}{R} = \cosh \frac{x}{R} \cosh \frac{y}{R} - \sinh \frac{x}{R} \sinh \frac{y}{R} \cos \alpha,$$

from which, using $\cosh(x + y) = \cosh x \cosh y + \sinh x \sinh y$, we get

$$\cosh \frac{x+y}{R} - \cosh \frac{z}{R} = \sinh \frac{x}{R} \sinh \frac{y}{R} (1 + \cos \alpha).$$

If $R \to \infty$, then, similarly to Euclidean case, we get $\varphi = x + y - z = \Omega(1)$.

On the other hand, if $R \to 0$, we get $\varphi = \Omega(R)$. Note that $D_{min} \geq |z - 2| + \frac{|x-1|+|y-1|}{(n-1)/2} = |\varepsilon + \delta - \varphi| + \frac{|\varepsilon|+|\delta|}{(n-1)/2}$. This gives us a lower bound $D_{min} = \Omega(1/n)$ in spherical and Euclidean spaces and $D_{min} = \Omega(\min(R, 1)/n) = \Omega\left(\frac{1}{n \max(\sqrt{-c}, 1)}\right)$ in hyperbolic space. From this and Equation 9 the bound on $D(S_n)$ follows.

Now, let us get an upper bound on optimal distortion $D_{opt}(S_n)$. To do this, we explicitly construct an embedding with sufficiently low distortion $D(S_n)$.

Let $v$ be the central node, then we spread all other nodes uniformly on a 2-dimensional circle of radius 1 centred at $v$. The smallest angle between two points is $2\pi/n$. Therefore, from the law of cosines, the distance between leaves is at least $k$ with

$$\cosh \frac{k}{R} = 1 + \left(1 - \cos \frac{2\pi}{n}\right) \sinh^2 \frac{1}{R}.$$

Note that for any two leaves $v_i$ and $v_j$ we have that $d(f(v_i), f(v_j)) \leq 2$. In particular, the closer two leaves are, the greater the difference $2 - d(f(v_i), f(v_j))$ is. Hence, the distance between adjacent leaves is the worst case and thus $D_{opt}(S_n)$ can be upper bounded as

$$D_{opt}(S_n) \leq \frac{\binom{n}{2}}{2\binom{n+1}{2}} \left(2 - R \cdot \operatorname{arccosh}\left(\left(1 - \cos\frac{2\pi}{n}\right)\sinh^2\frac{1}{R} + 1\right)\right)$$

$$= \frac{(n-1)}{2(n+1)} \left(2 - R \cdot \operatorname{arccosh}\left(\left(1 - \cos\frac{2\pi}{n}\right)\sinh^2\frac{1}{R} + 1\right)\right)$$

Note that $1 - \cos\frac{2\pi}{n} = \Theta\left(\frac{1}{n}\right)$ and $\sinh^2\left(\frac{1}{R}\right) = \Theta\left(e^{2/R}\right)$. Then, $\operatorname{arccosh}\left(\Theta\left(\frac{1}{n}e^{2/R}\right) + 1\right)$ behaves as $\sqrt{2e^{2/R}/n}$ if $2/R \ll \log n$ and as $\frac{2}{R} - \log n$ if $2/R \gg \log n$. Therefore, we get

$$D_{opt}(S_n) = O\left(R \log n\right) = O\left(\frac{\log n}{\sqrt{-c}}\right).$$

### B.4 PROOF OF THEOREM 4.4 (DISTORTION-BASED CURVATURE FOR COMPLETE GRAPHS)

If $d = n - 2$, then we are given a $(n-1)$-simplex, which can be embedded into $n - 2$-dimensional spherical space. Indeed, the radius of circumscribed hypersphere for the $(n-1)$-simplex with side length $a$ is known to be $R = a\sqrt{\frac{n-1}{2n}}$. Since we want the *spherical* distance between all points to be equal to one, we need to choose $a$ accordingly:

$$\sin\frac{\alpha}{2} = \frac{a}{2R} \quad \text{for} \quad \alpha = \frac{1}{R}.$$

Here $\alpha$ corresponds to the angle giving the arc length 1, while the condition on $\sin\frac{\alpha}{2}$ relates $\alpha$ and $a$ since $\alpha$ is the angle in a triangle with side lengths $\alpha, R, R$. This implies that

$$2R \arcsin\frac{a}{2R} = 1,$$

and solving this equation for $a$ yields

$$a = \sqrt{\frac{2n}{(n-1)}} \frac{1}{2\arcsin\sqrt{\frac{n}{2(n-1)}}}.$$

Plugging this back into the formula for the radius $R$ we obtain

$$R = \frac{1}{2\arcsin\sqrt{\frac{n}{2(n-1)}}},$$

so that we have

$$C_{n-2}^{dist}(K_n) = 4\left(\arcsin\sqrt{\frac{n}{2(n-1)}}\right)^2.$$

Finally, let us show that if $c \to -\infty$, then optimal distortion $D_{opt}(K_n) \to 0$. This result follows from the fact that $D_{opt}(S_n) \to 0$, because to embed a clique, it is sufficient to embed a star on $n+1$ node with edge lengths 1/2 and then remove the central node.

### B.5 PROOF OF THEOREM 4.2 (THRESHOLD-BASED CURVATURE FOR STAR)

We show below that for any $n$, any dimension $d$ and some curvature $c$ there exists a perfect embedding. Therefore $C_d^{dist}(S_n)$ consists of curvatures for which such perfect embedding exists.

First, let us note that if there exists a perfect embedding $f$ for some curvature $c$, then there exists a perfect embedding for any curvature $c' < c$. Indeed, w.l.o.g., we assume that the central node $v$ is mapped to the origin of a hyperspherical coordinate system and other points $v_1, \ldots, v_n$ can be described by their radii and angles. We know that the distance between $v$ and any $v_i$ is at most 1 and the distance between any pair $v_i, v_j$ is larger than one. Now we change curvature to $c' < c$ and

keep hyperspherical coordinates the same. Then the distance between $v$ and any $v_i$ does not change, while the distance between nodes $v_i, v_j$ increases.

Now, it is easy to see that $C$ increases with $d$: if there exists an embedding to some dimension $d$, then, obviously, the same embedding works for $d' > d$. Further, $C$ decreases with $n$ since if there exists embedding of $S_n$, then we can easily construct an embedding of $S_{n'}$ for $n' < n$ by removing some nodes.

Now, let us construct a perfect embedding of $S_n$ and estimate the required curvature. Recall that in a perfect embedding all leaves have to be inside the ball of radius 1 around the central node $v$ and also the distance between any two leaves has to be larger than one.

It is easy to see that if we managed to spread $n$ points inside the ball of radius 1 with distances more than 1 between them, then we can move each point along the radius up to distance 1 from $v$ preserving this property. Therefore, it is sufficient to spread all points on a hypersphere.

Let us first consider $d = 2$. In this case we can find an explicit analytic expression for $C$. This will be an upper bound for any $d > 2$.

First, assume that $n > 6$. In this case we have to consider only hyperbolic space, since $n$ neighbors would not fit to a circle of radius 1 in neither spherical or Euclidean spaces.

We will find such largest curvature $C$ which allows to have distance exactly 1 between the closest leaves. In this case we cannot embed $S_n$ in a space of curvature $C$, but can embed in a space of any smaller curvature. We use Equation 7 and let $\alpha = \frac{2\pi}{n}$:

$$\cosh \frac{1}{2R} = \frac{1}{2 \sin \frac{\alpha}{2}},$$

$$R = \frac{1}{2 \operatorname{arccosh} \frac{1}{2 \sin \frac{\alpha}{2}}},$$

$$C = -\left(2 \operatorname{arccosh} \frac{1}{2 \sin \frac{\alpha}{2}}\right)^2.$$

Note that if $n$ is large, then $\sin \frac{\alpha}{2} = \sin \frac{\pi}{n-1} \sim \frac{\pi}{n-1}$. Then, $\operatorname{arccosh} \frac{1}{2 \sin \frac{\alpha}{2}} \sim \operatorname{arccosh} \frac{n-1}{2\pi} \sim \log n$, so we get $C \sim -4 \log^2 n$.

Now, let us consider $n \le 6$. Obviously, for $n = 6$ we have $C = 0$.

If $n < 6$, then $C > 0$. We use Equation 8:

$$\cos \frac{1}{R} = \frac{\cos \alpha}{1 - \cos \alpha},$$

$$C = \left(\arccos \frac{\cos \alpha}{1 - \cos \alpha}\right)^2.$$

### B.6  Proof of Theorem 4.3 (threshold-based curvature for trees)

As for $S_n$, we prove that for $T_b$ a perfect embedding exists for $d = 2$ and some curvature $c$. Then, similarly to the previous section, it is clear that $C$ increases with $b$ and decreases with $n$.

For the lower bound on $C$, we have to guarantee that an embedding exists. For this, we provide the following construction (see Figure 5 for an illustration). (Below we assume that the curvature is large enough for our construction to work, then we estimate the required curvature.) First, we take the node $v$ and consider a circle of radius 1 around this node. We spread $b + 1$ neighbors uniformly around this node. For our construction to work, we need all distances between these nodes to be larger than 1. Now, at some step of the algorithm, assume that we have all nodes at level $l$ placed at some circle centered at $v$ and all distances between the nodes at level $l$ are larger than 1. Our aim is then to find positions for all nodes at level $l + 1$.

Let us take any node at $l$-th level. Consider two points $u_l$ and $u_r$ on the same circle at distance 1 from the node $u$ to the left and to the right, respectively. Let $u, u_l, z_l$ and $u, u_r, z_r$ form equilateral

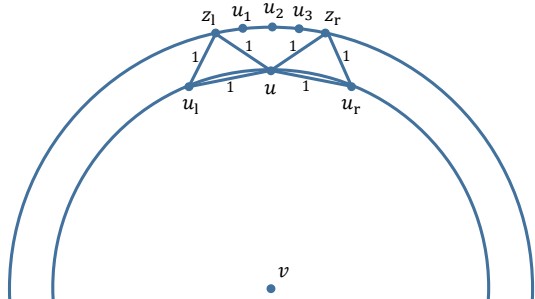

Figure 5: Threshold-based embedding of trees

triangles (with sides equal to 1). Then we let the points at $l+1$-th level to be spread on the circle centred at $v$ and passing through $z_l$ and $z_r$. The children of $u$ ($u_1, \ldots, u_b$) will be placed on the circular arc between $z_l$ and $z_r$. As usual, we want $u_i$ and $u_{i+1}$ to be at distance at least 1 from each other. Moreover, they have to be at distance at least 1 from children of other nodes. Also, note that placing $u_1, \ldots, u_b$ between $z_l$ and $z_r$ guarantees that these nodes are closer than 1 to their parent node $u$ but at the same time at a distance larger than 1 from all other nodes at $l$-th level. Also, all points at $l$-th level are far enough from points at $l+2$-th level.

It remains to find a maximum curvature such that the required conditions are satisfied. Let $r$ and $r'$ be radii of circles at $l$-th and $l+1$-th levels and let $2\alpha = \angle u_l v u$. We know (the law of cosines and $\cos 2\alpha = 1 - 2\sin^2 \alpha$) that

$$\cosh \frac{1}{R} = 1 + 2\sinh \frac{r}{R} \sin^2 \alpha. \tag{10}$$

And the only condition we need for the whole procedure to work is that we have enough space on the circular arc for placing $b$ nodes there:

$$\cosh \frac{1}{R} \leq 1 + 2\sinh \frac{r'}{R} \sin^2 \frac{\alpha}{b}.$$

Now we note that $\sin^2 \frac{\alpha}{b} \geq \frac{\sin^2 \alpha}{b^2}$ for all $b \geq 1$. Therefore, it is sufficient to have

$$\cosh \frac{1}{R} \leq 1 + 2\sinh \frac{r'}{R} \cdot \frac{\sin^2 \alpha}{b^2}. \tag{11}$$

Combining Equation 10 and Equation 11, we obtain:

$$\sinh \frac{r'}{R} \geq b^2 \cdot \sinh \frac{r}{R}.$$

To achieve this, it is sufficient to have:

$$\frac{r' - r}{R} \geq 2\log b,$$

$$R \leq \frac{r' - r}{2\log b}.$$

It remains to find the lower bound for $r' - r$ and it is easy to see that $r' - r$ decreases with $r$. Therefore, it is sufficient to consider only the second step of the construction procedure, when we move from the circle of radius 1 to the next one. In this case, $r = 1$ and

$$r' = 2\operatorname{arccosh} \frac{\cosh 1}{\cosh 1/2}.$$

So, we have

$$-C = \frac{1}{R^2} \leq \left( \frac{2\log b}{2\operatorname{arccosh} \frac{\cosh 1}{\cosh 1/2} - 1} \right)^2.$$

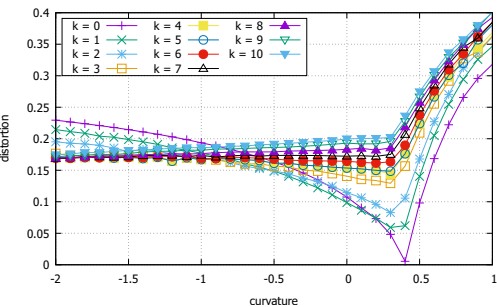

Figure 6: Adding $k$ edges to one node of a cycle on 10 nodes

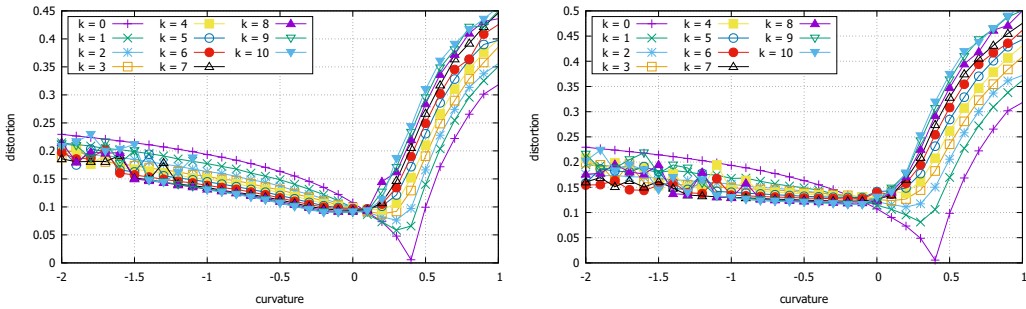

Figure 7: Adding one (left) or two (right) dangling edges to $k$ subsequent nodes of a cycle on 10 nodes

## C  COMBINATIONS OF SIMPLE GRAPHS

In this section, in order to get additional intuition, we illustrate how combining several simple subgraphs may affect the optimal curvature. We choose distortion for this illustrative example, since for this loss we theoretically obtained non-trivial results: for some simple graphs the optimal curvature is negative, while for other it is positive.

Indeed, for trees (and, in particular, stars) the optimal curvature is minus infinity and for cycles the optimal curvature is positive. Our intuition is that combining stars and cycles in one graph would give optimal curvature located between the curvatures of these simple graph.

Indeed, we observe such expected behavior for two illustrative examples shown in Figures 6 and 7. In Figure 6 we take a cycle of length 10 and iteratively add edges to one of its nodes, i.e., we create a star on one node. In Figure 7 we take the same cycle and add small stars sequentially to several of its nodes: in the left figure we just add one dangling edge for each node (i.e., create a star subgraph on 4 nodes), in the right figure we add two edges. In all cases, the optimal curvature drifts from a positive one for a cycle to negative values. This drift is more pronounced In Figure 6, where we create just one (relatively) large star.

## D  VOLUME-BASED CURVATURE APPROXIMATION

In this section, we describe the proposed curvature approximation. An input parameter of this estimator is the maximum neighborhood size $k_{max}$ (we fix $k_{max} = 3$ in the experiments). We take a node $v$ and for any $k \leq k_{max}$ we compute the number of nodes $N_k(v)$ at distance exactly $k$ from it. Then, we discount this value using the number of edges between the nodes. This captures our intuition that we need enough space to place the neighbors, but the connected neighbors are not required to be far away from each other. Formally, let $E_k(v)$ be the number of non-edges between the pairs of nodes at distance $k$. Then, we say that $N_k'(v)$ is such that $\binom{N_k'(v)}{2} = E_k(v)$. Finally, we compute the curvature $c_k(v)$ such that the area of a hypersphere or radius $k$ is equal to $N_k'(v)$.

Table 3: Dataset description

| Dataset | num. nodes | num. edges |
|---|---|---|
| Karate club (Zachary, 1977) | 34 | 78 |
| Dolphins (Lusseau et al., 2003) | 62 | 159 |
| Football (Newman & Girvan, 2004) | 115 | 613 |
| Political books (Newman, 2006) | 105 | 441 |
| Conflict (Ward et al., 2007) | 127 | 253 |
| Chicago (Eash et al., 1979) | 822 | 821 |
| CSPhDs (W. De Nooy & Batagelj, 2011) | 1025 | 1043 |
| Euroroad (Šubelj & Bajec, 2011) | 1039 | 1305 |
| EuroSiS (Khokhar, 2015) | 1272 | 6424 |
| Power (Watts & Strogatz, 1998) | 4941 | 6594 |
| Facebook (Leskovec & Mcauley, 2012) | 4039 | 88234 |

And our estimate for $v$ is $c(v) = \min_{k \leq k_{max}} c_k(v)$ (the worst-case curvature). To get a curvature of the whole graph, we take a median of all $c_k(v)$. (Similarly to other curvature estimators, one can consider a subsample of nodes instead of the whole node set to speed up the computation).

## E EXPERIMENTS

### E.1 DATASETS

Datasets used in our experiments are listed in Table 3. In all cases, we take only the giant connected component (numbers of nodes and edges in Table 3 correspond to already filtered dataset). Synthetic datasets are described in Section 4.3. The dataset $T_{3,6}$ is a tree with branching factor 3 and depth 6.

### E.2 THRESHOLD-BASED LOSS FUNCTIONS

While our theoretical results in Section 4 hold for any threshold-based loss function, we noticed that for less trivial datasets, where the perfect embedding is impossible, the choice of the loss function becomes important. In most of the experiments we use zero-one loss function:

$$\frac{1}{\binom{n}{2}} \sum_{i,j} I[v_i \sim v_j] \cdot I[d(f(v_i)f(v_j)) > 1] + (1 - I[v_i \sim v_j]) \cdot I[d(f(v_i), f(v_j)) \leq 1]$$

which essentially measures how well the values $I[d(f(v_i)f(v_j)) > 1]$ approximate $I[v_i \sim v_j]$. However, due to the fact that the real-world graphs are usually sparse, optimizing zero-one loss may lead to algorithms biased towards sparse embeddings. A standard solution to this is to consider balanced accuracy instead of accuracy, which leads to balanced zero-one loss:

$$\frac{\sum_{i,j} I[v_i \sim v_j] \cdot I[d(f(v_i)f(v_j)) > 1]}{2 \sum_{i,j} I[v_i \sim v_j]} + \frac{\sum_{i,j} (1 - I[v_i \sim v_j]) \cdot I[d(f(v_i), f(v_j)) \leq 1]}{2 \sum_{i,j} (1 - I[v_i \sim v_j])}.$$

The recent study by Gösgens et al. (2019) analyses the problem of bias discussed above in detail. It formally defines and analyzes a *constant baseline* requirement and shows that it is satisfied, in particular, by Pearson correlation coefficient between the vectors consisting of $I[v_i \sim v_j]$ and $I[d(f(v_i)f(v_j)) > 1]$. So, we add the corresponding Pearson loss (one minus correlation coefficient) to our experiments (see (Gösgens et al., 2019)).

### E.3 EXPERIMENTAL SETUP

Our experiments are based on a publicly available implementation of graph embedding by Gu et al. (2019).[8] The main advantage of this algorithm is that it works for any curvature: negative, positive or zero.

---

[8]https://github.com/HazyResearch/hyperbolics

Table 4: Compare curvature estimators, distortion, synthetic datasets

| | Ollivier | | Forman | | Avg. sectional | | Grad. descent | |
|---|---|---|---|---|---|---|---|---|
| Dataset/dim | c | loss | c | loss | c | loss | c | loss |
| $S_{100}$ / 2 | 0.0 | 0.334 | -97.0 | **0.058** | -1.0 | 0.304 | -1.45 | 0.289 |
| $S_{100}$ / 10 | 0.0 | 0.127 | -97.0 | **0.014** | -1.0 | 0.1 | -30.2 | 0.025 |
| $T_{3,6}$ / 2 | -0.33 | **0.136** | -2.0 | 0.254 | -0.5 | 0.174 | -9.77 | 0.271 |
| $T_{3,6}$ / 10 | -0.33 | **0.016** | -2.0 | 0.243 | -0.5 | 0.036 | 0.0 | 0.077 |
| $K_{100}$ / 2 | 0.99 | 0.348 | 100 | 0.841 | 0.125 | **0.342** | -48.7 | **0.163** |
| $K_{100}$ / 10 | 0.99 | 0.134 | 100 | 0.841 | 0.125 | **0.124** | -305 | **0.015** |
| $C_{100}$ / 2 | 0.0 | **0.106** | 0.0 | **0.106** | 0.01 | 0.257 | 0.0 | **0.106** |
| $C_{100}$ / 10 | 0.0 | **0.106** | 0.0 | **0.106** | 0.01 | 0.255 | 0.0 | **0.106** |
| $K_{100,100}$ / 2 | 0.0 | 0.372 | -196 | **0.294** | 0.008 | 0.372 | 2.1 | 0.309 |
| $K_{100,100}$ / 10 | 0.0 | 0.301 | -196 | **0.275** | 0.008 | 0.301 | -139 | **0.27** |

Table 5: Compare curvature estimators, zero-one loss, synthetic datasets

| | Ollivier | | Forman | | Avg. sectional | | Grad. descent | |
|---|---|---|---|---|---|---|---|---|
| Dataset | c | loss | c | loss | c | loss | c | loss |
| $S_{100}$ / 2 | 0.0 | 0.02 | -97.0 | **0.004** | -1.0 | 0.027 | 0.0 | 0.02 |
| $T_{3,6}$ / 2 | -0.33 | 0.006 | -2.0 | **0.005** | -0.5 | **0.005** | 0.0 | **0.002** |
| $C_{100}$ / 2 | 0.0 | **0.005** | 0.0 | **0.005** | 0.01 | 0.009 | 0.16 | 0.012 |
| $K_{100,100}$ / 2 | 0.0 | 0.496 | -196 | **0.492** | 0.008 | 0.496 | -531 | 0.493 |
| $K_{100,100}$ / 10 | 0.0 | 0.489 | -196 | 0.491 | 0.008 | **0.480** | -124 | 0.490 |

We modified the implementation for our task: in particular, we added a regime responsible for minimizing threshold-based loss functions. E.g., zero-one loss is proportional to

$$\sum_{i,j} I[v_i \sim v_j] \cdot I[d(f(v_i)f(v_j)) > 1] + (1 - I[v_i \sim v_j]) \cdot I[d(f(v_i), f(v_j)) \leq 1].$$

This function is not differentiable, so we replace it by gradient descent friendly function which we call ReLuLoss:

$$\sum_{i,j} I[v_i \sim v_j] \cdot \mathrm{ReLu}(d(f(v_i), f(v_j)) - (1-\varepsilon)) + (1 - I[v_i \sim v_j]) \cdot \mathrm{ReLu}((1+\varepsilon) - d(f(v_i), f(v_j)),$$

where $\mathrm{ReLu}(x) = \max(0, x), \varepsilon = 0.001$. In our experiments we observed that such loss function outperforms other analogues (e.g., MSE and sigmoid-based smoothing) since it speeds up the convergence. To adapt to various threshold-based loss functions, we also reserve several iterations to gradient steps based on the corresponding sigmoid-based smoothing.

Note that in experiments we use only the more advanced Forman curvature $\hat{F}(G)$. For computing Ollivier and Forman curvature we use the open source package by Ni et al. (2015).[9]

For the embedding algorithm, in all experiments we fix *epochs = 1000* and we also try several learning rates (usually from 0.01 to $10^4$). We noticed that learning rate significantly affect the performance of the learning algorithm and also the optimal learning rate is usually smaller for smaller datasets and it is larger for spaces of negative curvature. We also use the option *use_adagrad = True* since we noticed that it inproves the embedding into Euclidean spaces (otherwise there is a peak in distortion at $c = 0$).

Let us emphasize that in our experiments we do not care about overfitting, because we do not analyze the embedding algorithm, our aim is to find a curvature which minimizes loss function. (It can be also thought of as though we focus on graph reconstruction task).

---

[9]`https://github.com/saibalmars/GraphRicciCurvature`

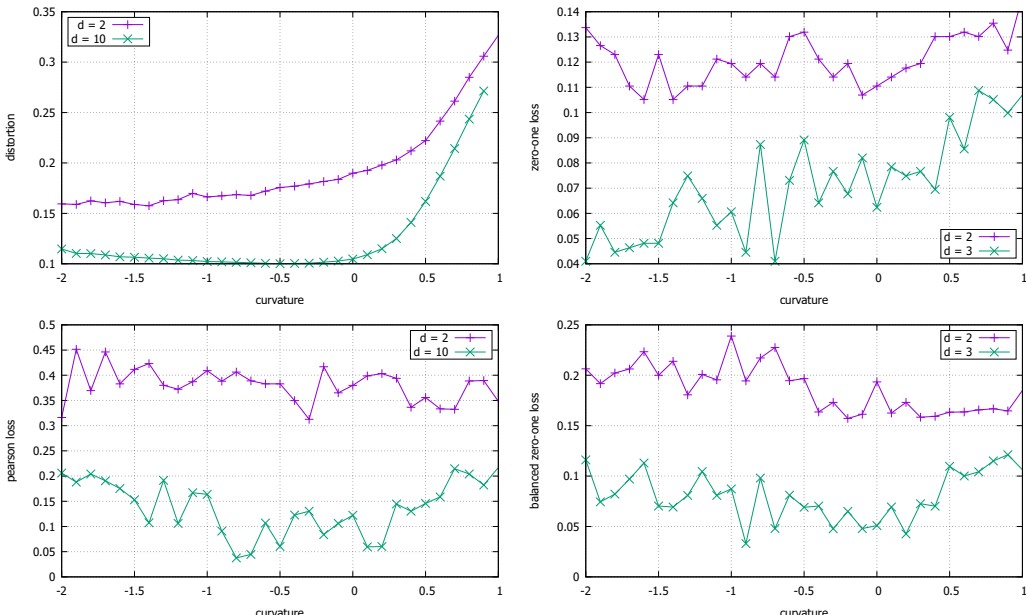

Figure 8: Effect of curvature, Karate club dataset

To plot all the figures we try curvatures in some range (usually between -2 and 1) with step size 0.1. Our code is publicly available.[10]

### E.4 COMPARISON OF CURVATURES ON SYNTHETIC DATASETS

See the results in Tables 4 and 5.

### E.5 EFFECT OF DIMENSION AND LOSS FUNCTION

In this section, we visualize the dependence of the loss function on curvature (see Figures 8-18). In particular, we demonstrate that the optimal curvature significantly depends on both loss function and dimension. More precisely, while our theoretical results hold for any threshold-based loss function, for less trivial graphs the optimal curvature depends on the particular choice.[11]

The obtained results support our intuition (based on theoretical results) that while hyperbolic space helps in many cases for small dimensions, it becomes less pronounced in larger dimensions. This is most clearly seen for distortion loss (see, e.g., Karate club, Football, CSPhDs, Euroroad, EuroSiS, Power).

### E.6 EMPIRICAL ANALYSIS OF VOLUME-BASED CURVATURE

As discussed in the previous section, optimizing distortion leads to more stable results than threshold-based losses, so we focus on this loss here. It turns out that the proposed volume-based estimator is able to reflect well whether it is reasonable to embed a network to a hyperbolic space for a given loss function if we care about distortion. Table 6 shows the estimated values of a curvature.[12]

---

[10]https://drive.google.com/open?id=12i5LD6yTyFRDrgkMJEencDAI0Au6Isrn

[11]We note that the curves corresponding to threshold-based losses are noisy, especially for small datasets, since such loss functions are discrete and multimodal, so they are harder to optimize. To smooth the results, we added several tricks to the optimizer and also choose the optimum embedding based on several random restarts of the algorithm. Although the results are still noisy, the general trends are usually clearly seen.

[12]The value 0.1 appears due to a technical reason: in spherical space of large curvature there can be no points at distance $k$, so we limit considered curvatures.

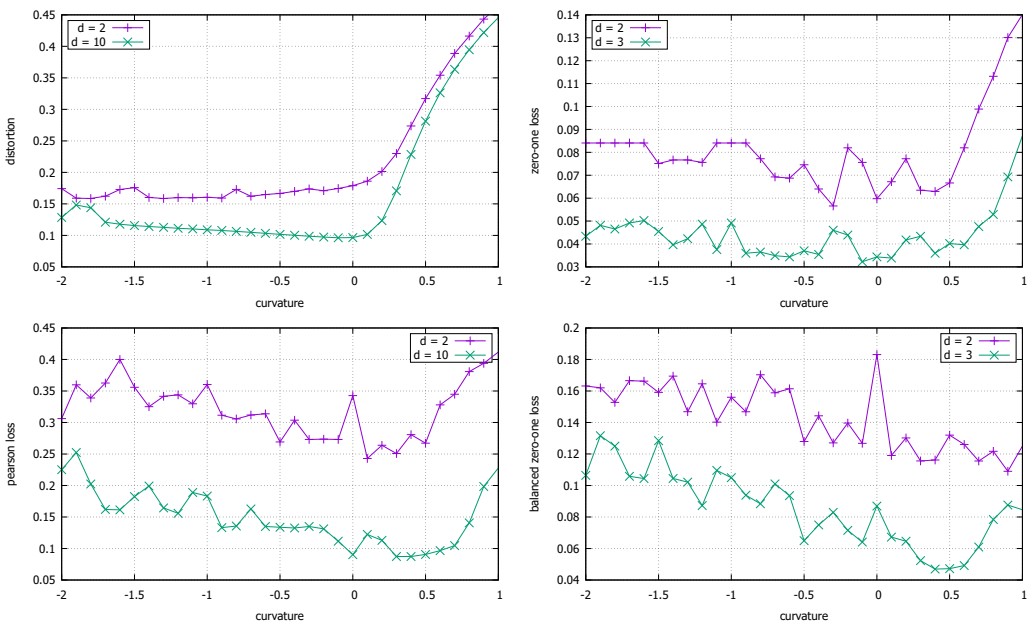

Figure 9: Effect of curvature, Dolphins dataset

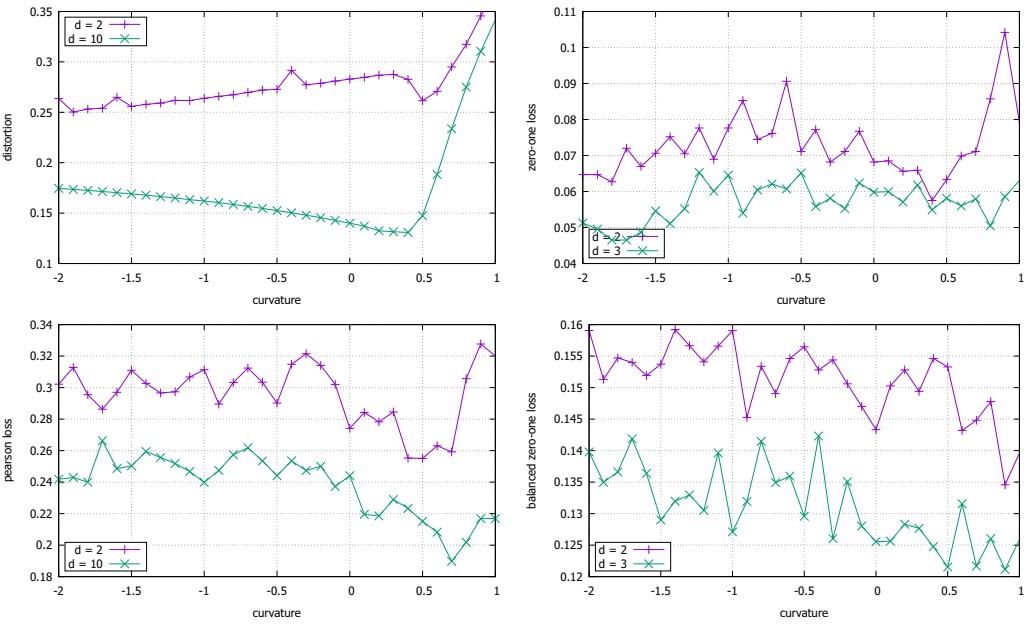

Figure 10: Effect of curvature, Football dataset

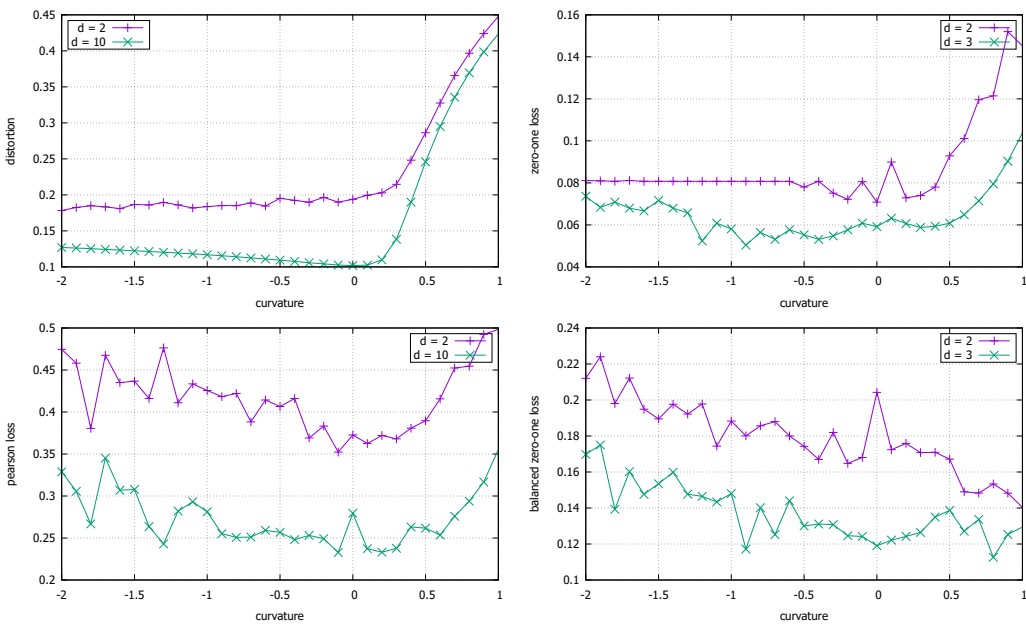

Figure 11: Effect of curvature, Political books dataset

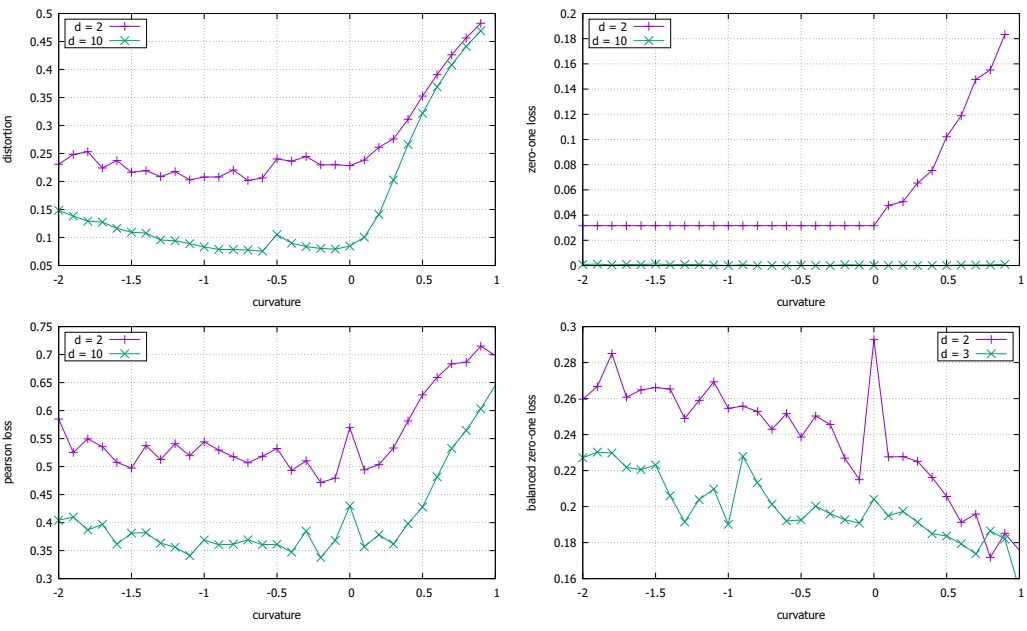

Figure 12: Effect of curvature, Conflict dataset

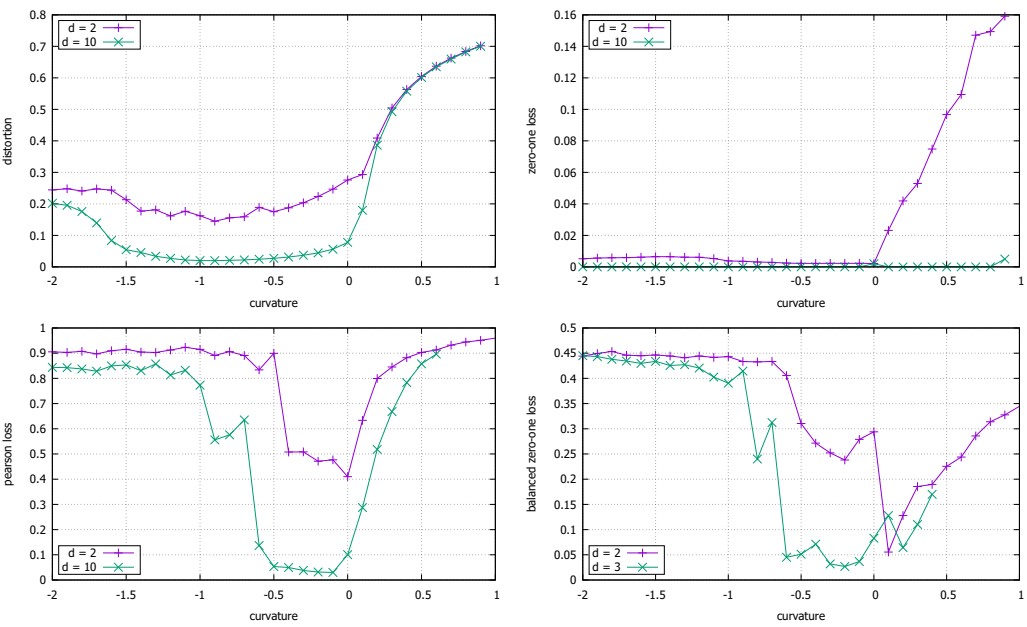

Figure 13: Effect of curvature, Chicago dataset

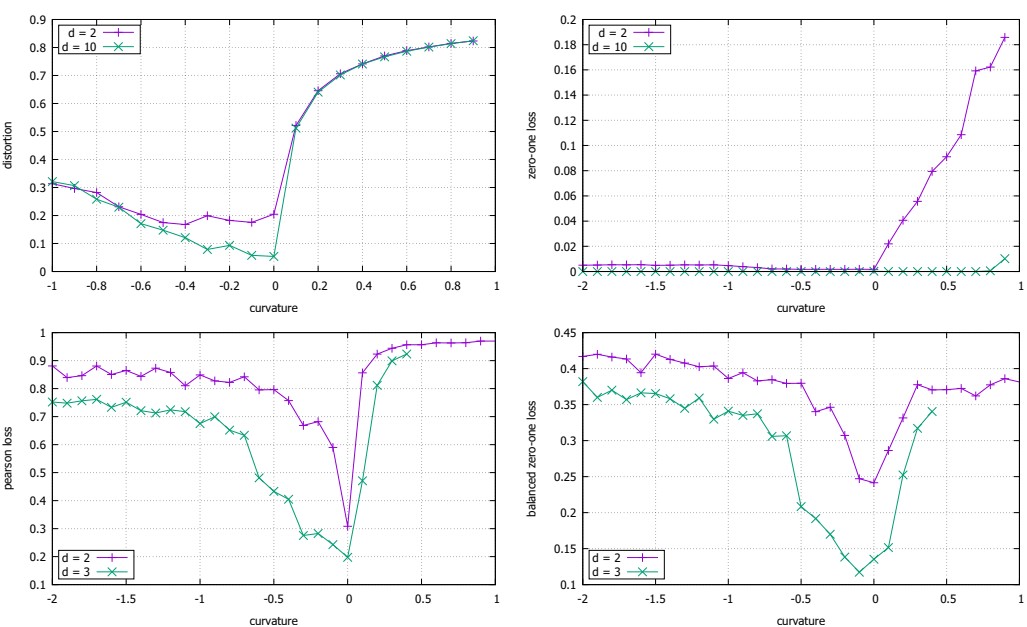

Figure 14: Effect of curvature, CSPhDs dataset

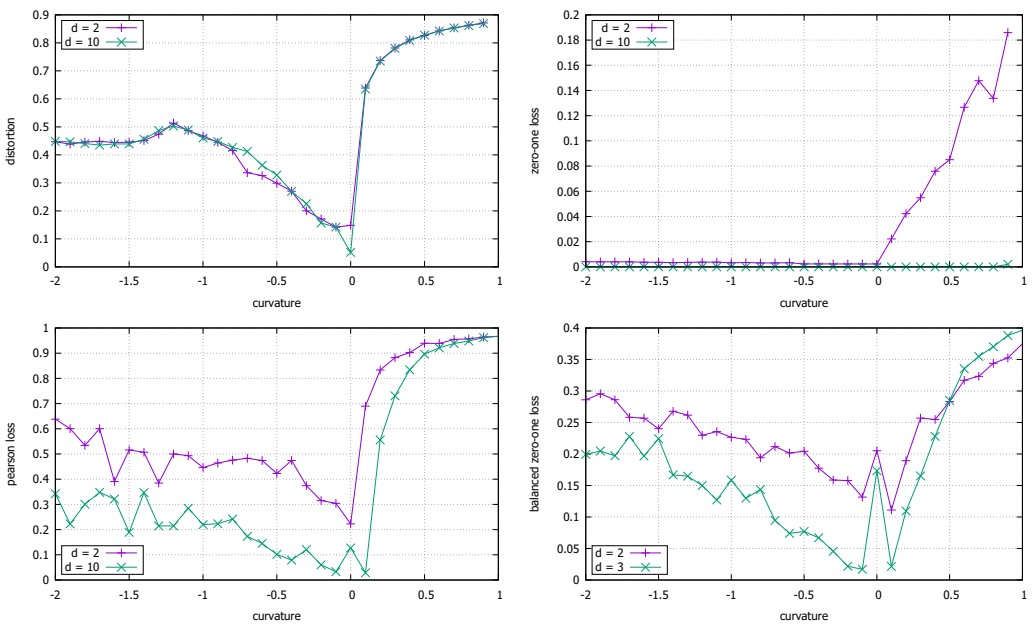

Figure 15: Effect of curvature, Euroroad dataset

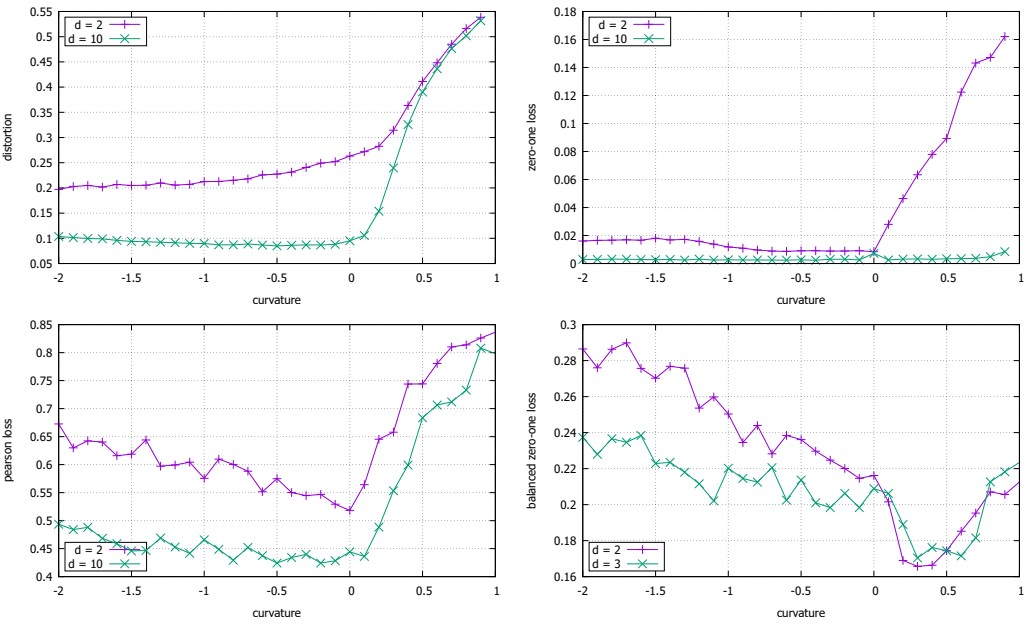

Figure 16: Effect of curvature, EuroSiS dataset

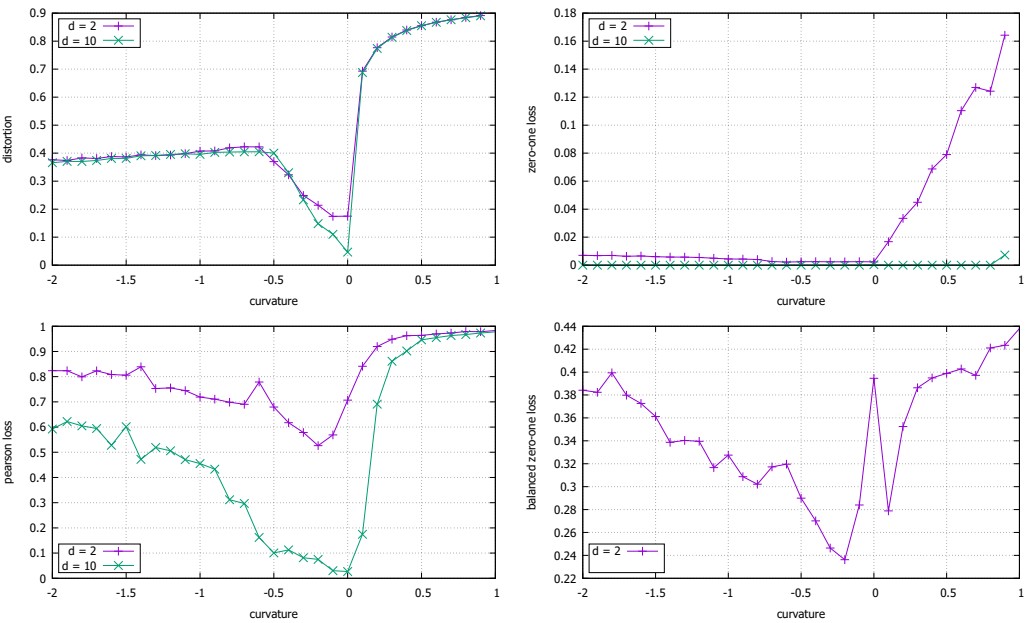

Figure 17: Effect of curvature, Power dataset

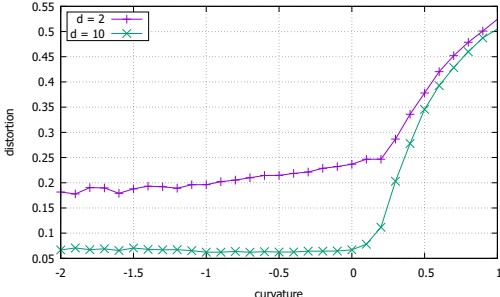

Figure 18: Effect of curvature, Facebook dataset, distortion loss

Table 6: Volume-based curvature

| Dataset | $d = 2$ | $d = 10$ |
|---|---|---|
| Karate club | -6.7 | 0.1 |
| Dolphins | -3.3 | 0.1 |
| Football | -2.1 | 0.1 |
| Political books | -1.6 | 0.1 |
| Conflict | -1.8 | 0.1 |
| Chicago | 0.1 | 0.1 |
| CSPhDs | 0.07 | 0.1 |
| Euroroad | 0.1 | 0.1 |
| EuroSiS | -3.7 | 0.1 |
| Power | 0.1 | 0.1 |
| Facebook | -11 | 0.1 |

One can see that for $d = 2$ in many cases significantly negative curvature is predicted which is indeed confirmed by our experiments, shown in Figures 8-18 (distortion loss). Non-negative curvature is predicted for Chicago, CSPhDs, Euroroad and Power and indeed for these datasets the optimal curvature is close to zero. In contrast, for $d = 10$ negative curvature is never predicted, which agrees with the experiment well. So, in these cases it is not reasonable to embed networks to a hyperbolic space.

