# OpenReview forum: "Global graph curvature"
_ICLR.cc/2020/Conference — Reject_

### Official Review · AnonReviewer2 · 2019-10-23
**Official Blind Review #2**

**Rating:** 6

**Review:**

Summary:
This paper is about curvature as a general concept for embeddings and ML applications. The origin of this idea is that various researchers have studied embedding graphs into non-Euclidean spaces. Euclidean space is flat (it has zero curvature), while non-Euclidean spaces have different curvature; e.g., hyperbolic space has a constant negative curvature. It was noted that trees don't embed well into flat space, while they embed arbitrarily well into hyperbolic space.

All of the notions of curvature, however, are defined for continuous spaces, and have to be matched in some sense to a discrete notion that applies to graphs beyond a particular class like trees. The authors study this setting, consider a variety of existing notions of curvature for graphs, introduce a notion of global curvature for the entire graph, and now to efficiently compute it. They also consider allowing these concepts to vary with the downstream task, as represented by the loss function.


Pros, Cons, and Recommendation

The study of the various proposed distances is fairly interesting, although it's hard to say what the takeaway here is. I think the part I'm struggling with the most is the motivation. Why do we care about using a space of constant curvature? True, we do so when it's appropriate---we embed trees into hyperbolic space. But when we have a more complicated and less regular graph, then compressing all of that curvature information into a scalar doesn't seem like a good idea, and indeed that's the point of the Gu et al work that's being built on here: it mixes and matches various component spaces that each have constant curvature, but altogether have varying curvature.

At the same time, I like the idea of studying a bunch of proposed measures and attempting to gain new insights. This is a pretty unusual paper for ICLR, since the experimental section is really barely there, and what's most interesting are really these atomic insights. If the authors work on the motivation I would consider accepting it---for now I gave it weak accept.



Comments:
- The distinction between what the authors think of as "global" and "local" curvatures is confusing and should be explained further. From what I can see, the authors think of global as being a scalar, and local as being defined at each point; intuitively these seem like pretty bad labels. I would think of the "global" one as being coarse, and the "local" as being more refined, since it contains a lot more information. This is also related to the motivation: why stuff all of this information into one single scalar curvature? It forces you to take averages, while Gu et al defined a distribution over the local curvatures.

If the idea is to simply use one space and not a product, there are in fact various spaces with non-constant curvature, e.g., the complex manifold CH^n.

- Why do your need your graph to be unweighted at the very beginning of Section 2? On the other hand, you may want to define your graph to be connected for the distortion function to be well-defined.

- The statement "1, graph distances are hard to preserve:..." isn't really meaningful, since for the example in 4.3.1, it is possible to embed that graph arbitrarily well. That is, even if the distortion isn't 0, it can be made as small as we desire. There are indeed graphs that are hard to embed (i.e., have lower bounds that do not go to 0) in reasonably tractable spaces, and the authors actually prove such a result, but the star graph is not one of these.

- There's various tricks that actually make some of these graphs very easy to embed. One example is K_n in 4.33. Instead of just embedding K_n, embed the star graph on n+1 nodes, and place a weight of 1/2 on each edge. Now every pair of (non-central) nodes is at distance exactly. 1, and this thing is embeddable into hyperbolic space, etc. Interestingly, this is actually predicted by the Gromov hyperbolicity (for K_n_ that the authors briefly mention.

The reason I bring this up is that even if the authors' project is successful, simple graph transformations may induce much better embeddings. That's fine, though, although it should be mentioned.

- Can the authors write out what's going on for the hyperbolic lower bound on D_min in the  proof of Thm. 4.1?

**Experience Assessment:**

I have published one or two papers in this area.

**Review Assessment: Checking Correctness Of Derivations And Theory:**

I assessed the sensibility of the derivations and theory.

**Review Assessment: Checking Correctness Of Experiments:**

I assessed the sensibility of the experiments.

**Review Assessment: Thoroughness In Paper Reading:**

I read the paper at least twice and used my best judgement in assessing the paper.

---

> ### Author Response · Authors · 2019-11-08
> **To Reviewer 2: Motivation behind our research**
>
> 1) We chose spaces of constant curvature since such spaces are currently used in various applications (like, e.g., Poincare GloVe) and also relatively easy to implement and use in practice. While product space proposed by Gu et al. are able to achieve a superior quality, they are harder to implement and they require the signature (combination of spaces) to be chosen before the embedding (in their experiments, Gu et al. performed grid search to choose a combination of spaces to use).
>
> 2) Our approach can be used as a tool in some more advanced approaches. E.g., one could embed a graph into a space of constant curvature and then refine this embedding by embedding the residual in some other space of another constant curvature. Or different parts of the graph can be embedded into different spaces.
>
> 3) These constant curvature spaces are easier to understand and therefore to obtain results in, as we do in this paper. These results and insights can eventually be extrapolated to analyze curvature in other, more complicated spaces. As far as we know, the problem of analyzing different notions of curvature for embedding graphs (even in simple spaces) was not considered before, so our aim is to stimulate research in this direction.
> The motivation part will be improved in the revised version of the paper.

---

> ### Author Response · Authors · 2019-11-08
> **To Reviewer 2: Reply to comments and questions**
>
> Thank you for the careful reading and many valuable comments.
>
> “The distinction between what the authors think of as "global" and "local" curvatures is confusing and should be explained further. From what I can see, the authors think of global as being a scalar, and local as being defined at each point; intuitively these seem like pretty bad labels. I would think of the "global" one as being coarse, and the "local" as being more refined, since it contains a lot more information.“
>
> Our use of “global” and “local” is based on their uses in complex network analysis. Here a local property of a graph refers to a property that depends on some small neighborhood of a node, while a global property depends on the whole graph. An example is the local clustering coefficient, which computes the fraction of links between the neighbors of a given node, and the global clustering, which computes the fraction of triangles in the whole networks compared to the total number of paths of length two. Also note that both global and local properties can be scalars, vectors or even functions. So although in this case the global curvature is a scalar, this does not mean that any notion of global curvature has to be, nor that we only consider scalars as global properties.
>
> “If the idea is to simply use one space and not a product, there are in fact various spaces with non-constant curvature, e.g., the complex manifold CH^n.”
>
> Thank you for pointing us to these manifolds. We did consider other manifolds, such as the Bolza surface (compact manifold with constant negative curvature). However, computing distances on this manifold (which is obtained as a the factorization of the Poincare disc over some group) is non-trivial. Thus, to make sure we could efficiently implement the computations needed for our experiments we initially choose to stick with these three classes of manifolds. From a practical perspective, hyperbolic, euclidean and spherical spaces are already widely used and there are embedding techniques developed for them.
>
> “Why do your need your graph to be unweighted at the very beginning of Section 2?”
>
> We want it to be easier to define both distortion and threshold loss. If there is a weighted graph, then one has to convert this weight to a distance to compute distortion or to 0-1 to compute threshold loss. This can be done and the analysis can be extended, but this would add another dimension to the research, so we decided to start from unweighted and undirected graphs.
>
> “On the other hand, you may want to define your graph to be connected for the distortion function to be well-defined.”
>
> Thanks for pointing this out, we’ll add this assumption. Note that in practice it is reasonable to assume that a graph is connected since connected components can be embedded separately.
>
> “The statement "1, graph distances are hard to preserve:..." isn't really meaningful, since for the example in 4.3.1, it is possible to embed that graph arbitrarily well. That is, even if the distortion isn't 0, it can be made as small as we desire. There are indeed graphs that are hard to embed (i.e., have lower bounds that do not go to 0) in reasonably tractable spaces, and the authors actually prove such a result, but the star graph is not one of these.”
>
> Thank you, this motivation may indeed seem unclear. We wanted to show that the star with 4 nodes is an example of a very small graph, where only minus infinite curvature works. But minus infinity gives a degenerate tree-like structure and if a graph is not a tree, then it becomes a problem, as illustrated by our example with bipartite graphs. We will change this statement and make it more clear.
>
> “There's various tricks that actually make some of these graphs very easy to embed. One example is K_n in 4.33. Instead of just embedding K_n, embed the star graph on n+1 nodes, and place a weight of 1/2 on each edge. Now every pair of (non-central) nodes is at distance exactly. 1, and this thing is embeddable into hyperbolic space, etc. Interestingly, this is actually predicted by the Gromov hyperbolicity (for K_n_ that the authors briefly mention.”
>
> Indeed, in the proof of Theorem 4.4 (at the very end of section B.4, on page 15) we actually refer to this trick with converting K_n to star. This is the reason why cliques may have two minima - a positive one and minus infinity. We’ll mention this trick explicitly in the main text.
>
> “Can the authors write out what's going on for the hyperbolic lower bound on D_min in the  proof of Thm. 4.1?”
>
> Could you, please, clarify this question? Is this about the intuition or are there any particular transition which is unclear?

---

> > ### Comment · AnonReviewer2 · 2019-11-14
> > **Thanks!**
> >
> > Thanks for your answers to my questions. One response:
> >
> > - Could you, please, clarify this question? Is this about the intuition or are there any particular transition which is unclear?
> >
> > Originally I was confused about the expression near the bottom of Page 14, where D_min = \Omega(min(R,1)/n) in hyperbolic space. The initial reason for my confusion is that you should be able to drive this lower bound down to 0 for n a constant, but at first glance that formula doesn't appear to behave that way.
> >
> > In effect though, you're getting this by making your R go to 0 by increasing c. So it's behaving as expected.
> >
> > I should say one more thing for this part: although we have this nice exact distortion formula D(f), in the math literature on embeddings, most of the time the distortion is up to some constant factor, e.g., distortion(f) = min_c 1/c * D(f), or even just measured as the worst-case type of thing, max_{u,v} d(f(u),f(v))/d_G(u,v)  /   min_{u,v} d(f(u),f(v))/d_G(u,v).
> >
> > Under these definitions, I think the only lower bound is 0 for trees in hyperbolic space, even with R and n fixed.
> >
> > Anyway, this isn't directly relevant for your paper, but it might be good to comment on the different notions of distortion, since there's slightly different uses in other fields.

---

> > > ### Author Response · Authors · 2019-11-15
> > > **Revised paper**
> > >
> > > Thank you for the suggestions! We uploaded a revised paper, see our comment above https://openreview.net/forum?id=ByeDl1BYvH&noteId=SJelGQWnjH . In particular, we tried to improve the motivation part. We also added a comment on the different notions of distortion.

---

### Official Review · AnonReviewer3 · 2019-10-24
**Official Blind Review #3**

**Rating:** 6

**Review:**

The paper presents a novel notion named glocal graph curvature, which offers a solution to determine the optimal curvature for embedding. In particular, the global graph curvature depends on both dimension and loss function used for the network embedding. Besides, the authors studied the existing local curvatures and show that the existing graph curvatures may not be able to properly capture the global graph structure curvature. Extensive results demonstrate the statements proposed in the paper. In general, I like the paper due to its nice presentation, interesting view of graph curvature, and solid theoretical analysis. However, I am not familiar with graph curvature. All I can say is the approach is intuitively appealing, the text is well written and easy to follow, even for an outsider. I do not know any related works or what to expect from the results. I could not find anything wrong with this paper, but also do not have any intelligent questions to ask.

**Experience Assessment:**

I do not know much about this area.

**Review Assessment: Checking Correctness Of Derivations And Theory:**

I carefully checked the derivations and theory.

**Review Assessment: Checking Correctness Of Experiments:**

I carefully checked the experiments.

**Review Assessment: Thoroughness In Paper Reading:**

I read the paper at least twice and used my best judgement in assessing the paper.

---

> ### Author Response · Authors · 2019-11-15
> **Revised paper**
>
> Thank you for the feedback!
>
> We uploaded a revised paper, the improvements are listed in the comment above https://openreview.net/forum?id=ByeDl1BYvH&noteId=SJelGQWnjH

---

### Official Review · AnonReviewer4 · 2019-10-30
**Official Blind Review #4**

**Rating:** 3

**Review:**

This paper considers the problem of embedding graphs into continuous spaces.  The emphasis is on determining the correct dimension and curvature to minimize distortion or a threshold loss of the embedding.

Pros:
The problem is clearly stated and easy to understand.
The limitations of the three local curvatures are shown empirically and theoretically

Cons:
Experiments seem inconclusive, with no discussion of the results
Proposed global curvature characterizes the optimal embedding parameters but not a different, efficiently calculable discrete curvature to approximate them
Analysis of particular graph families doesn’t necessarily inform what to expect from embedding large graph data

Overall, I lean towards rejecting this paper.  The problem does seem an important one, but it seems the main contribution of this paper is comparing the local curvatures against an oracle for determining optimal curvature in embedding space, without putting forward an alternative method.

Questions:
Why is it reasonable to take these curvature metrics and use them directly as the curvature of the ambient space at all?  Especially given that Ollivier curvature belongs to a small interval and Forman curvature is always negative.

Does any of the graph family analysis carry over to more general graphs?  For example, assuming some priors about the appearance of these families as subgraphs, or the observed features of real networks in [1]?

[1] J. Leskovec, D. Chakrabarti, J. Kleinberg, C. Faloutsos, and Z. Gharamani. Kronecker graphs: an approach to modeling networks. arXiv:0812.4905v1, 2008.

**Experience Assessment:**

I have read many papers in this area.

**Review Assessment: Checking Correctness Of Derivations And Theory:**

I assessed the sensibility of the derivations and theory.

**Review Assessment: Checking Correctness Of Experiments:**

I assessed the sensibility of the experiments.

**Review Assessment: Thoroughness In Paper Reading:**

I read the paper at least twice and used my best judgement in assessing the paper.

---

> ### Author Response · Authors · 2019-11-08
> **To Reviewer 4: Reply to comments and questions**
>
> Indeed, the main contributions of this paper are theory and the acquired insights. In particular, we proved the limitation of all existing simple estimators. The main aim is to bring attention to the problem of curvature computation for embeddings and start research in this important direction. Note that we also propose a simple curvature estimator which has desired properties: it depends on dimension and designed for threshold-based loss.
>
> “Why is it reasonable to take these curvature metrics and use them directly as the curvature of the ambient space at all?  Especially given that Ollivier curvature belongs to a small interval and Forman curvature is always negative.”
>
> These curvatures are widely used in complex network analysis, so our aim was to test their applicability in practice. Indeed, Ollivier curvature has a limited interval and Forman curvature is often highly negative (but not always, see \hat{F} in Section 4.3.3). Additionally, we also consider a heuristic curvature that was actually used in practice. However, the main drawback of all these curvatures is the fact that they do not depend on dimension or loss function, which is crucial, as we show in this paper.
>
> “Does any of the graph family analysis carry over to more general graphs?  For example, assuming some priors about the appearance of these families as subgraphs, or the observed features of real networks in [1]?”
>
> This is an excellent question and definitely on our list of future projects. The difficulty is that it is not simply the joint appearance of families as subgraphs but also how they are related among each other. For example, it matters if a star has one peripheral node that belongs to a cycle where some of its nodes also belong to the star or none of them do. The real issue is that curvature is not simply a function of subgraph occurrences, but really a function of the intricate graph structure. We are currently working on the analysis of such graph combinations and will reply with more details when we get some insights.

---

> ### Author Response · Authors · 2019-11-15
> **Revised paper**
>
> Thank you for the feedback!
>
> We uploaded a revised paper, see our comment above https://openreview.net/forum?id=ByeDl1BYvH&noteId=SJelGQWnjH
>
> In particular, we addressed your question on more general graphs via additional experiments (Figures 6 and 7).
>
> We also improved and extended the experimental part and tried to better motivate the applicability of our research.

---

### Author Response · Authors · 2019-11-15
**Revised paper**

We would like to thank the reviewers for their comments and suggestions. We uploaded a revised version of the paper, where we tried to address all concerns.

We made the following updates:

- Improved the motivation part in the text as suggested by Reviewer 2 (for complete details see our reply to Reviewer 2).

- Let us remark that when starting the project we hoped that global curvature could be an intrinsic property of a network but as the result of our research we realized (and proved for some graphs) that it is also influenced by the properties of an ambient space, e.g. dimensionality. In the text we put additional emphasis on the important fact that usually a network which seems to be negatively curved (for some small dimension) becomes more neutral as dimensionality grows (which is confirmed by our theory and experiments). This means that in large dimensions hyperbolic embeddings may not be needed.

- Regarding the question of Reviewer 4 on combinations of simple graphs, we totally agree that it is an important question and, while it is extremely hard to give a complete answer theoretically, we did empirical simulations, added a section with an illustration of the results (Appendix C) and referred to this at the beginning of Section 4.3.

- We agree that experiments were a bit inconclusive and tried to improve this part. Now experiments show that 1) global curvature significantly depends on dimension and loss, 2) all existing curvatures are unable to capture the global one. More illustrations of these conclusions are in Appendix E.5. In Appendix E.6 we added a more detailed analysis of the proposed volume-based estimator, where we show that it is able to capture well whether the space is negatively curved and it also predicts the behavior of optimal curvature as dimension grows.

- To support our conclusion that the optimal curvature depends on a loss function, we also considered more threshold-based loss functions including the correlation coefficient which is (in some sense) unbiased, as discussed in a very recent paper arXiv:1911.04773. While our theoretical results hold for all loss functions, for more complex graphs the optimal curvature may indeed depend on a particular threshold-base loss.

- We also made some other smaller changes, as we promised in our reply to Reviewer 2.

We plan to make a small update of the paper later today (by adding some illustrative figures to Appendix E.5).

---

### Decision · Program_Chairs · 2019-12-19

**Decision:**

Reject

**Comment:**

This paper studies the problem of embedding graphs into continuous spaces.  The authors focus on determining the correct dimension and curvature to minimize distortion or a threshold loss of the embedding. The authors  consider a variety of existing notions of curvature for graphs, introduce a notion of global curvature for the entire graph, and how to efficiently compute it.

Reviewers were positive about the problem under study, but agreed that the current manuscript somewhat lacks a clear contribution. They also pointed out that the goal of using a global notion of curvature should be better motivated. For these reasons, the AC recommends rejection at this time.